# The Abstraction Gap in Vision-Language Causal Reasoning

**Chinh Hoang** [1]   **Mohammad Rashedul Hasan** [1]

## Abstract

Vision-language models (VLMs) generate fluent causal explanations, but current evaluations cannot distinguish linguistic plausibility from faithful causal reasoning. We introduce a dual-probe methodology that isolates these properties. The Text-Only Probe measures linguistic quality. The Chain-Text Probe requires models to first generate explicit causal chains. The Abstraction Gap (AG) metric quantifies the normalized performance difference. Evaluating eight VLMs on CAGE (Causal Abstraction Gap Evaluation), a benchmark of 49,500 questions across 5,500 images spanning Pearl's causal hierarchy, we find seven models exhibit AG exceeding 0.50 with text scores of 6–8 but chain scores below 2.5. Fine-tuning on 45,000 chain-annotated examples fails to close the gap. However, one model achieves near-zero AG. The capability exists within current VLM architectures and depends on pretraining and architectural choices. CAGE provides a diagnostic tool for assessing faithful causal reasoning in VLMs.

## 1. Introduction

Vision-language models (VLMs) have achieved impressive performance across visual understanding tasks, from recognizing objects and their attributes to generating fluent explanations about complex visual scenarios (Liu et al., 2023a; Alayrac et al., 2022; Li et al., 2023a; Chen et al., 2024c; Wang et al., 2024b; Hong et al., 2024). When presented with a photograph of a beach scene (Figure 1), these models can accurately identify the number and color of umbrellas, infer the time of day from shadows, and answer sophisticated questions about hypothetical changes, for example, "What would happen if strong winds began blowing from the mountains?" Their responses are often coherent and

[1]Department of Electrical and Computer Engineering, University of Nebraska–Lincoln, Lincoln, Nebraska, USA. Correspondence to: Mohammad Rashedul Hasan <hasan@unl.edu>.

*Proceedings of the $43^{rd}$ International Conference on Machine Learning*, Seoul, South Korea. PMLR 306, 2026. Copyright 2026 by the author(s).

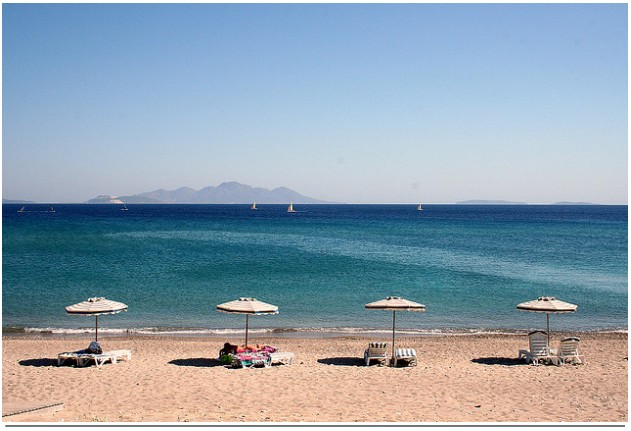

| Level | Question | Causal Chain | Answer |
|---|---|---|---|
| 1 | What is the color of the leftmost beach umbrella? | N/A | White and Orange. |
| 2 | If a strong wind began to blow from the direction of the mountains, what would happen to the beach umbrellas? | Strong wind → Force on umbrellas → Umbrellas topple or fly away | The umbrellas may topple or fly away. |
| 3 | Had this photo been taken during a holiday weekend instead of what appears to be a quiet day, how would the beach scene differ? | Holiday weekend → More visitors → Crowded beach → Fewer visible umbrellas | The beach would be more crowded with fewer visible umbrellas. |

*Figure 1.* Can VLMs reason causally, or do they generate plausible language without structural understanding? CAGE tests this through questions at Pearl's three levels (association, intervention, counterfactual). For intervention and counterfactual questions (Levels 2-3), models must first generate a lightweight causal chain (e.g., 'Strong wind → Force on umbrellas → Umbrellas topple') before providing text. Most VLMs answer fluently but fail to generate these minimal chains, exposing a severe Abstraction Gap between linguistic plausibility and structural reasoning.

contextually appropriate. The convincing causal language implies an ability to reason about causal relationships and alternative scenarios. However, a central question remains. Do these models genuinely understand underlying causal mechanisms in visual scenes, or are they generating plausible causal language through statistical pattern matching? As VLMs move into high-stakes applications such as medical imaging and autonomous driving, this question takes on practical urgency.

Recent work probes whether VLMs possess such structural understanding by testing their ability to work with causal

chains, a format that externalizes causal dependencies into explicit sequential relationships (Zhang et al., 2025b). In these benchmarks, models select valid causal chains from pre-provided candidates and achieve up to 68% accuracy. However, this verification task provides only a partial test of the generativity property described above. Models may succeed through pattern matching or surface-level discrimination without having internalized causal structure sufficiently to construct it independently. A more complete test would assess whether models can generate causal abstractions without scaffolding. Research on human cognition suggests conceptual structure precedes linguistic expression (Levelt, 1989; Jackendoff, 2002), with speakers constructing a preverbal representation of relationships before formulating language. By analogy, VLMs with genuine causal understanding should be able to generate structural representations before verbalizing explanations.

The distinction reflects a broader challenge in explainable AI of distinguishing between *plausibility* and *faithfulness* in model explanations (Lyu et al., 2024; Agarwal et al., 2024). Plausible explanations seem logical and convincing to humans and inspire user trust through linguistic fluency. Faithful explanations accurately represent the model's internal reasoning process and provide genuine insight into how decisions are made. Recent work argues these properties can diverge and warns that pressures to generate user-friendly explanations may increase plausibility without improving faithfulness (Agarwal et al., 2024). Empirical studies find the faithfulness of large language model (LLM) self-explanations is not guaranteed and varies by task, model, and explanation type (Madsen et al., 2024). The **plausibility-faithfulness gap** poses particular risks in high-stakes domains where understanding the reasoning behind a decision is as consequential as the decision. For VLMs deployed in such contexts, evaluating faithfulness becomes necessary. Evidence shows the gap manifests broadly. VLMs confidently offer plausible explanations while ignoring contradictory visual evidence (Guan et al., 2024) and generate hallucinated explanations for straightforward visual questions (Tong et al., 2024). They also rely on spurious correlations instead of genuine visual understanding (Yang et al., 2026; Zečević et al., 2023).

We investigate the faithfulness of VLM explanations by examining their causal reasoning capabilities. Cognitive science research motivates this focus. Genuine causal understanding relies on internal generative models that represent latent causes and support structured physical reasoning (Friston, 2010; Battaglia et al., 2013). A defining property of such internal models is their **generativity**, which is the capacity to construct structural representations of causal relationships. A model with genuine causal understanding should be able to abstract its reasoning into a structural form. To assess whether VLMs possess this capability, we

ground our investigation in Judea Pearl's Ladder of Causation (Pearl & Mackenzie, 2018; Pearl, 2009), which defines three ascending levels, from association (observing correlations) through intervention (reasoning about action effects) to counterfactual (imagining alternative scenarios). Each successive level requires deeper causal understanding.

We test this structural generativity through a minimal sufficiency condition that examines whether models can identify important causal elements in visual scenes and represent their relationships using simple arrow notation. For an intervention question about wind affecting beach umbrellas, this abstraction would be "strong wind $\rightarrow$ force on umbrellas $\rightarrow$ umbrellas topple." Chains at this level of simplicity require only identifying relevant entities and their sequential dependencies. Failure to generate such minimal structures would indicate that linguistic plausibility masks a lack of structural understanding, even when models provide fluent textual explanations of the same causal relationships. We define the **Abstraction Gap** metric to quantify this disparity by measuring the difference between text generation performance and chain generation performance.

For large-scale evaluation of the Abstraction Gap, we construct **CAGE** (Causal Abstraction Gap Evaluation), a benchmark comprising 49,500 open-ended questions across 5,500 COCO images (Lin et al., 2014)[1]. CAGE spans all three levels of Pearl's causal hierarchy and requires generation without scaffolding. We evaluate eight open-source VLMs using a multi-judge framework with human validation on 4,500 responses to ensure evaluation reliability beyond any single annotator's biases.

Our methodological contribution is a *dual-probe evaluation framework* designed to isolate plausibility from faithfulness. The *Text-Only Probe* sets baseline performance using standard textual responses and measures plausibility. The *Chain-Text Probe* requires models to first generate a lightweight causal chain before providing textual answers and measures faithfulness through structural abstraction. Prompts request chains first, followed by text explanations to mirror the cognitive sequence of structure-before-expression. The inability to generate chains while delivering fluent text exposes the plausibility-faithfulness gap. Chain requirements are minimal, with one causal link required for intervention questions and two for counterfactual questions. The methodology differs from existing causal reasoning benchmarks that implicitly address causal aspects or focus on multiple-choice formats with predefined structures. Figure 1 illustrates an example from CAGE showing questions at three causal levels and the lightweight chain format required for Levels 2–3.

Our evaluation uncovers a severe Abstraction Gap across

---

[1]The benchmark is available on Kaggle.

most current VLMs. While models achieve strong text-based performance (averaging 6–8 out of 10), they score below 2.5 out of 10 on chain generation and often yield blank or malformed outputs. The disparity persists across model scales (7B to 76B parameters) and architectural families. Apparent causal reasoning in text does not guarantee the ability to extract and represent causal structure. However, one model achieves near-zero Abstraction Gap, which confirms that causal chain generation is possible within current VLMs. We examine the training factors correlating with this outcome in Section 4.2.

Can explicit supervision close this gap? We investigate this through causal instruction fine-tuning on 45,000 chain-annotated examples. Despite substantial training data, the Abstraction Gap persists for models with low baseline performance. Text quality improves modestly, but chain generation shows limited improvement. Fluent text generation constitutes a linguistic skill where VLMs excel, while structural abstraction requires capabilities that current training frameworks do not adequately support.

We observe a related dissociation in visual grounding. Hallucination benchmarks test *perceptual grounding* (Sun et al., 2024; Li et al., 2023b), whether generated content corresponds to objects present in images. The Abstraction Gap tests *structural grounding*, the degree to which causal language reflects relational understanding. Our analysis shows these capabilities are independent. Fine-tuning on causal chains does not affect hallucination rates. Models trained for hallucination mitigation still exhibit severe Abstraction Gap. VLMs maintain separate mechanisms for representing what exists versus how things relate.

Our contributions include:

I. **CAGE Benchmark.** The first large-scale visual causal reasoning benchmark that requires explicit structural output generation, with 49,500 open-ended questions across Pearl's three causal levels grounded in real-world COCO images.

II. **Dual-Probe Evaluation Framework.** A methodology that isolates abstraction capability from linguistic fluency, validated through multi-judge evaluation and human agreement studies.

III. **Discovery of the Abstraction Gap.** Systematic evaluation of eight VLMs that exposes a severe gap between linguistic fluency and structured reasoning, with analysis of training factors associated with reduced Abstraction Gap.

IV. **Training Framework Analysis.** Fine-tuning experiments on 45,000 chain-annotated examples show explicit chain supervision does not reduce the Abstraction Gap for most models. We analyze training factors linked to low Abstraction Gap in successful cases.

## 2. Related Work

**Causal Reasoning Benchmarks.** We position CAGE against existing visual causal reasoning benchmarks. CausalVLBench (Komanduri et al., 2025) uses synthetic images with known causal structures for binary and prediction tasks grounded in Pearl's hierarchy. CELLO (Chen et al., 2024a) provides predefined causal chains and evaluates reasoning through multiple-choice selection. Info-CausalQA (Ka et al., 2025), TimeCausality (Wang et al., 2025), CausalVQA (Foss et al., 2025), and MuCR (Li et al., 2025) test causal understanding through selection-based formats across infographics, temporal image pairs, video, and synthetic image pairs, respectively. These benchmarks assess whether models can verify or select valid causal relationships. CAGE addresses the complementary question of whether models can *generate* causal structures independently. High-stakes deployment requires models to construct explanations de novo. Plausibility alone does not guarantee faithful reasoning. MM-CoT (Zhang et al., 2025b) shows VLMs achieve 68% accuracy on chain verification. Our evaluation finds comparable models score below 2.5/10 on chain generation. Selection performance overestimates structural generation capability.

**Training Methods and Grounding.** Contrastive methods (Patel et al., 2024) and counterfactual fine-tuning (Zhang et al., 2026) improve compositional reasoning but prioritize continuous embeddings for discrimination. Our experiments show that the Abstraction Gap persists across diverse architectures and training frameworks. We also find that structural grounding (causal abstraction) and perceptual grounding (object hallucination (Li et al., 2023b; Sun et al., 2024)) are independent capabilities. Models trained for hallucination mitigation still exhibit severe AG.

Appendix A.1 provides detailed coverage of benchmarks, training methods, and theoretical foundations.

## 3. Methodology

### 3.1. Problem Formulation

We formalize the evaluation of causal reasoning in VLMs. Let $\mathcal{I}$ denote an image space and $\mathcal{Q}$ a space of natural language questions. A VLM $M : \mathcal{I} \times \mathcal{Q} \rightarrow \mathcal{R}$ maps an image-question pair to a response. We examine two response modalities, textual explanations $r_t$ and causal chain structures $r_c$.

Following Pearl's Ladder of Causation (Pearl & Mackenzie, 2018; Pearl, 2009), we partition questions into three levels. **Level 1 (Association)** addresses observable correlations and properties. **Level 2 (Intervention)** involves hypothetical actions and their effects. **Level 3 (Counterfactual)** considers alternative scenarios contradicting observed reality. Each

level requires deeper causal understanding.

The plausibility-faithfulness gap predicts that text generation capability diverges from structural abstraction capability. Our goal is to quantify this divergence through a dual-probe methodology.

## 3.2. Dual-Probe Evaluation Framework

We design two complementary evaluation protocols to isolate plausibility from faithfulness.

*Text-Only Probe (Experiment A).* Given an image and a question, this probe elicits a textual response without structural constraints. The probe measures *plausibility*, the model's ability to generate linguistically fluent causal explanations.

*Chain-Text Probe (Experiment B).* For Level 2 and 3 questions, this probe requires the model to first generate a causal chain, then provide a textual response. Chains use arrow notation to connect causal elements ($e_1 \rightarrow e_2 \rightarrow \cdots \rightarrow e_n$), where each element denotes a cause or effect and each arrow encodes a causal dependency. The prompt requests the chain before text to mirror the cognitive sequence of structure-before-expression (Levelt, 1989; Jackendoff, 2002). The probe measures *faithfulness*, whether the model can abstract causal structure before verbalizing it. For Level 1 questions (association), only text responses are collected since no causal chain is required.

The ordering affects interpretation. A model with internal structural representations should generate causal structure first, then verbalize coherently. The inability to generate chains while producing fluent text exposes the plausibility-faithfulness gap.

**The Abstraction Gap Metric.** We measure the plausibility-faithfulness gap through the Abstraction Gap (AG), which quantifies the disparity between text and chain generation performance.

**Definition 3.1** (Abstraction Gap). Let $M$ denote a VLM, $\mathcal{D}_\ell$ a dataset at Pearl Level $\ell \in \{2, 3\}$, and let $T(M, d)$ and $C(M, d)$ denote evaluation scores (on a 0–10 scale) for text responses and causal chain generation, respectively, for each instance $d \in \mathcal{D}_\ell$. The Abstraction Gap is:

$$\mathrm{AG}_\ell(M) = \frac{\mathbb{E}_{d \in \mathcal{D}_\ell}[T(M, d)] - \mathbb{E}_{d \in \mathcal{D}_\ell}[C(M, d)]}{\mathbb{E}_{d \in \mathcal{D}_\ell}[T(M, d)]} \quad (1)$$

AG measures the relative reduction in performance when models generate explicit causal chains compared to text-only responses. Values near 1 indicate severe disparity. Such models offer plausible causal text but cannot generate valid structures. Values near 0 indicate aligned capabilities. Negative values (rare in practice) would indicate stronger structural than linguistic performance. Since AG quantifies

relative disparity, it should be interpreted alongside absolute scores. A model with low performance on both probes may exhibit low AG without possessing strong causal reasoning.

## 3.3. The CAGE Benchmark

We construct CAGE to measure the Abstraction Gap at scale through this dual-probe methodology.

*Dataset Composition.* CAGE comprises 5,500 images from the COCO dataset (Lin et al., 2014) with 49,500 question-answer pairs (nine questions per image, three per Pearl level). We draw 5,000 images from the COCO 2017 training set. These serve as training data for fine-tuning experiments. An additional 500 images from the COCO validation set form a held-out test set used for baseline evaluation under our dual-probe methodology, or for evaluating fine-tuned models to ensure no data leakage between training and post-fine-tuning evaluation.

*Chain Specifications.* We impose minimal structural requirements to ensure that failure cannot be attributed to representational complexity. Level 2 questions require at least one causal link ($|c| \geq 2$ elements) and Level 3 questions require at least two sequential links ($|c| \geq 3$ elements).

*Data Generation and Validation.* We employ GPT-4o (OpenAI, 2024) to generate questions, ground-truth answers, and causal chains, with detailed definitions of Pearl's hierarchy and well-formed examples for each level (Appendices A.2 and A.3). Each question is generated based on the image and its five associated COCO captions to ensure visual grounding. All content undergoes three-stage validation. First, level verification uses an LLM to classify each question's target causal level against Pearl's hierarchy. Second, visual answerability review of 600 randomly sampled questions confirms they are answerable from the image. Third, chain consistency review of 600 causal chains verifies logical coherence and sufficient causal links. All manual expert reviews achieve over 95% approval rate.

## 3.4. Evaluation Protocol

*Multi-Judge Framework.* To address potential annotation-evaluation circularity, we employ two independent LLM evaluators, GPT-4o (OpenAI, 2024) and Claude 3.5 Sonnet (Anthropic, 2024). Cross-judge agreement indicates evaluation reliability beyond single-annotator bias. See Appendix A.4 for evaluation prompts.

*Human Validation.* Expert annotators evaluate a stratified sample covering all question types and both probe conditions. The sample provides a gold standard for validating automated evaluation. Sample size and model selection are detailed in Section 4.

*Scoring Criteria.* Text responses are scored on accuracy, relevance, and logical consistency. Causal chains are

scored on structural validity, minimum link requirements, and causal coherence. All scores use a 0–10 scale. We report mean scores with 95% bootstrap confidence intervals (1,000 samples) and assess statistical significance using paired Wilcoxon signed-rank tests ($p < 0.05$).

### 3.5. Causal Instruction Fine-tuning

To investigate whether explicit supervision can close the Abstraction Gap, we fine-tune representative VLMs on the 45,000-example training set (5,000 images $\times$ 9 questions). Models are trained using standard autoregressive next-token prediction on sequences containing the image, question, causal chain (for Level 2–3), and text response. Through the causal attention mechanism, text generation is naturally conditioned on the preceding chain tokens. Substantial exposure to chain-formatted outputs should reduce the Abstraction Gap if the gap stems from limited training data. Persistence of the gap despite such exposure would suggest explicit chain supervision alone cannot instill structural abstraction capability. Fine-tuned models are evaluated on the held-out 500-image test set using the same dual-probe methodology.

## 4. Experiments

### 4.1. Experimental Setup

**Evaluated Models.** We evaluate eight open-source VLMs spanning diverse architectures, scales, and training objectives. *Large-scale* models include InternVL2 76B (Chen et al., 2024c) and CogVLM2 19B (Hong et al., 2024). *Mid-scale* models (11–13B) include LLaVA-NeXT 13B (Liu et al., 2024b), LLaVA-RLHF 13B (Sun et al., 2024), MiniGPT-4 13B (Zhu et al., 2024), and LLaVA-CoT 11B (Xu et al., 2025). *Smaller-scale* models (7–8B) include mPLUG-Owl2 8.2B (Ye et al., 2024) and Qwen-VL-Chat 7B (Bai et al., 2023).

The selection spans 7B to 76B parameters and includes models with specialized training objectives. LLaVA-RLHF targets hallucination mitigation, LLaVA-CoT targets chain-of-thought reasoning, and CogVLM2 employs a visual expert architecture. Five models (LLaVA-NeXT, mPLUG-Owl2, Qwen-VL-Chat, MiniGPT-4, LLaVA-CoT) are selected for fine-tuning experiments based on diverse baseline capabilities.

**Implementation.** All experiments use $8\times$ NVIDIA A40 GPUs (48GB each). Fine-tuning hyperparameters, optimizer configurations, and architecture-specific adaptation strategies are detailed in Appendix A.10.

**Evaluation Protocol.** We apply the dual-judge framework described in Section 3. All results include 95% bootstrap confidence intervals (1,000 samples). Statistical significance is assessed using paired Wilcoxon signed-rank tests, with exact $p$-values reported. Complete statistical tables with

per-judge breakdown are in Appendices A.5 and A.6.

### 4.2. Baseline Results

**Text-Only Probe: Measuring Plausibility.** Table 1 presents text response scores under the Text-Only Probe, which measures linguistic plausibility without structural constraints. Most VLMs achieve strong performance. CogVLM2, LLaVA-NeXT, Qwen-VL-Chat, and InternVL2 exceed 7.1 on Levels 2 and 3, with narrow 95% confidence intervals (width $\sim$0.20).

*Table 1.* Text-Only Probe (Experiment A): Average scores (0–10) on 500 COCO images, averaged across judges and over 3 questions per level. Blank responses scored as 0. *High blank rates: mPLUG-Owl2 (249), LLaVA-RLHF (2,467). Per-judge breakdown in Appendix A.5.

| Model | L1 | L2 | L3 |
|---|---|---|---|
| CogVLM2 | 8.38 | 8.26 | 8.11 |
| LLaVA-NeXT | 7.95 | 8.08 | 7.94 |
| MiniGPT-4 | 7.10 | 7.14 | 6.68 |
| mPLUG-Owl2* | 6.80 | 7.09 | 7.06 |
| Qwen-VL-Chat | 8.23 | 8.19 | 7.99 |
| InternVL2 | 7.97 | 7.57 | 7.12 |
| LLaVA-RLHF* | 5.12 | 6.76 | 6.92 |
| LLaVA-CoT | 7.92 | 3.07 | 4.68 |

Cross-judge agreement on relative model rankings is high, though absolute scoring tendencies differ slightly (Claude 3.5 assigns higher Level 1 scores, GPT-4o higher on Levels 2–3). LLaVA-CoT exhibits anomalously low Level 2 and 3 scores (3.07–4.68). The model struggles with causal questions even in a text-only format. These results show that most current VLMs generate linguistically plausible causal explanations when evaluated on text quality alone.

**Chain-Text Probe: Measuring Faithfulness.** Table 2 presents scores when models generate causal chains before text responses. The probe measures faithfulness, whether models can abstract causal structure from visual input. See Appendix A.7 for qualitative examples from this probe.

*Table 2.* Chain-Text Probe (Experiment B): Average scores (0–10) on 500 COCO images, averaged across judges. For L2–L3, scores are reported as (Text, **Chain**). All text-chain differences significant at $p < 0.001$. *High blank rates: CogVLM2 (532), LLaVA-RLHF (2,467).

| Model | L1 | L2 (T, C) | L3 (T, C) |
|---|---|---|---|
| CogVLM2* | 8.15 | (5.56, **0.54**) | (5.30, **1.37**) |
| LLaVA-NeXT | 7.71 | (7.96, **7.83**) | (7.64, **7.61**) |
| MiniGPT-4 | 7.06 | (5.14, **2.09**) | (4.91, **2.05**) |
| mPLUG-Owl2 | 6.65 | (5.77, **0.42**) | (5.84, **0.75**) |
| Qwen-VL-Chat | 8.01 | (6.86, **3.77**) | (6.76, **4.28**) |
| InternVL2 | 7.77 | (7.45, **3.90**) | (7.20, **2.51**) |
| LLaVA-RLHF* | 5.11 | (1.33, **0.81**) | (1.89, **1.32**) |
| LLaVA-CoT | 7.96 | (6.78, **0.97**) | (6.81, **1.60**) |

**The Abstraction Gap.** The preceding results indicate a clear pattern. While most VLMs generate fluent causal text

(scoring 6–8 in Experiment A), they struggle to represent the same relationships structurally (scoring below 2.5 on chains in Experiment B). Figure 2 quantifies this disparity through the Abstraction Gap (AG) computed according to Definition 3.1. Table 2a reports AG values by Pearl level and judge. Figure 2b ranks models by mean AG, from LLaVA-NeXT (0.03) to mPLUG-Owl2 (0.92). The gap is severe and statistically significant across most models ($p < 0.001$, Wilcoxon signed-rank test, Appendix A.6).

Seven of eight models exhibit AG values exceeding 0.50, meaning chain generation scores are less than half of text scores. CogVLM2 (AG = 0.88), mPLUG-Owl2 (0.92), and LLaVA-RLHF (0.84) represent severe gaps where models generate almost no valid chains despite strong text performance. MiniGPT-4 (0.70), LLaVA-CoT (0.67), and InternVL2 (0.57) show high gaps, while Qwen-VL-Chat yields a moderate gap (0.50). Comparing raw scores across Tables 1 and 2 makes this disparity concrete. CogVLM2 achieves a Level 3 text score of 8.11 but only 1.37 for chain generation, yielding AG $\approx$ 0.88. The pattern persists across judges and causal levels.

The wide variance in AG values (0.03 to 0.92) under the same rubric suggests that the metric captures capability differences, not task difficulty. A harder chain rubric would cause all models to score low. Instead, LLaVA-NeXT achieves high chain scores (7.61–7.83) while others approach zero under identical conditions.

High AG values reflect the plausibility-faithfulness gap. Models generate linguistically plausible causal explanations but cannot abstract the same relationships into explicit structural form. The inability to generate minimal causal chains (requiring only arrow notation) exposes that fluent text generation masks absent structural understanding.

**Variance Across Model Families.** The distribution of AG scores provides insight into what training factors correlate with causal generation capability. AG values range from 0.03 (LLaVA-NeXT) to 0.92 (mPLUG-Owl2), with seven of eight models exceeding 0.50. The variance does not align with model scale. InternVL2-76B exhibits high AG (0.57) despite being the largest model evaluated. It also does not align with the architectural family. Within the LLaVA family, AG ranges from 0.03 (LLaVA-NeXT) to 0.84 (LLaVA-RLHF). We explain the variance across the LLaVA family in Section 5.

**Blank Outputs.** For some models, such as LLaVA-RLHF, blank outputs primarily stem from the models' failure in structural abstraction. To distinguish whether these blanks arise from a lack of causal and structural reasoning or from harness-related issues, we analyze the model responses using a failure mode taxonomy. Table 3 reports the failure mode taxonomy of some models.

| Model | GPT-4o | | Claude 3.5 | |
|---|---|---|---|---|
| | L2 | L3 | L2 | L3 |
| CogVLM2 | 0.989 | 0.883 | 0.878 | 0.774 |
| LLaVA-NeXT | 0.041 | 0.105 | 0.022 | -0.028 |
| MiniGPT-4 | 0.783 | 0.774 | 0.631 | 0.612 |
| mPLUG-Owl2 | 0.969 | 0.933 | 0.911 | 0.853 |
| Qwen-VL-Chat | 0.590 | 0.491 | 0.487 | 0.437 |
| InternVL2 | 0.501 | 0.676 | 0.468 | 0.616 |
| LLaVA-RLHF | 0.928 | 0.850 | 0.828 | 0.764 |
| LLaVA-CoT | 0.737 | 0.711 | 0.635 | 0.604 |

*(a)* Numerical AG values by level and judge.

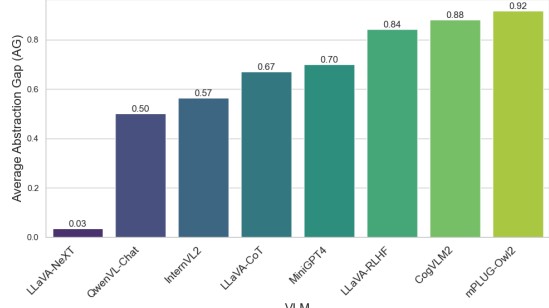

*(b)* Average AG across levels and judges.

*Figure 2.* Abstraction Gap across eight VLMs. (a) AG values per model, level, and judge. (b) Mean AG sorted from lowest (LLaVA-NeXT) to highest (mPLUG-Owl2).

*Table 3.* Failure mode taxonomy across models.

| Model | Format-Compliant | Blank | Text-Only | Other |
|---|---|---|---|---|
| LLaVA-NeXT | 89.0% | 0.0% | 1.1% | 9.9% |
| LLaVA-RLHF | 9.0% | 76.4% | 11.4% | 3.3% |
| CogVLM2 | 1.6% | 16.7% | 81.3% | 0.4% |
| MiniGPT-4 | 10.9% | 0.0% | 61.1% | 28.0% |

Blank rates vary dramatically under identical experimental conditions. LLaVA-RLHF generates blanks in 76.4% of cases, while LLaVA-NeXT generates none. If blanks were caused by harness issues, similar rates would be expected across models. The concentration of blanks in LLaVA-RLHF implicates its alignment training, which may induce output suppression when the model encounters unfamiliar structural or causal formats.

**Human Validation.** To validate automated evaluation, expert annotators assess 125 randomly sampled images (4,500 responses across LLaVA-NeXT and MiniGPT-4, covering both probe types). Table 4 presents human ratings, which closely mirror automated outcomes.

*Table 4.* Human evaluation (0–10) on 125 images. For Exp B L2–L3, scores are reported as (Text, **Chain**).

| Model | Exp A (Text) | | | Exp B (Text, Chain) | | |
|---|---|---|---|---|---|---|
| | L1 | L2 | L3 | L1 | L2 | L3 |
| LLaVA-NeXT | 8.94 | 9.02 | 8.80 | 9.07 | (8.82, **7.98**) | (8.49, **7.58**) |
| MiniGPT-4 | 7.91 | 8.21 | 7.63 | 7.90 | (6.52, **1.15**) | (6.02, **0.95**) |

Human evaluation validates the Abstraction Gap pattern.

MiniGPT-4 achieves a text score of 7.63 (Exp A, L3) but a chain score of only 0.95, while LLaVA-NeXT maintains strong performance on both (8.80 text, 7.58 chain). The consistency across automated and human evaluation supports our multi-judge framework.

## 4.3. Fine-tuning Results

We investigate whether explicit chain supervision can close the Abstraction Gap through causal instruction fine-tuning on 45,000 chain-annotated examples.

**Text-Only Probe After Fine-tuning.** Table 5 compares baseline and fine-tuned performance on text generation. Fine-tuning has varied impact across models. MiniGPT-4 exhibits substantial improvement ($p < 0.001$ for both judges), with Level 3 scores increasing from 6.68 to 7.78. LLaVA-NeXT and LLaVA-CoT show modest improvements that reach significance according to Claude but not GPT-4o. Qwen-VL-Chat represents mixed results with judges disagreeing on direction. Most concerning, mPLUG-Owl2 shows significant degradation ($p < 0.001$) with increased blank responses (from 249 to 1,095). Fine-tuning destabilizes the model's generation behavior.

*Table 5.* Text-Only Probe: Baseline (B) vs. Fine-tuned (F), averaged across judges. Bold indicates fine-tuned. *mPLUG-Owl2 blanks increase from 249 (B) to 1,095 (F).

| Model | L1 | L2 | L3 |
|---|---|---|---|
| LLaVA-NeXT (B) | 7.95 | 8.08 | 7.94 |
| **LLaVA-NeXT (F)** | **8.14** | **8.31** | **8.08** |
| MiniGPT-4 (B) | 7.10 | 7.14 | 6.68 |
| **MiniGPT-4 (F)** | **7.59** | **7.95** | **7.78** |
| mPLUG-Owl2 (B)* | 6.80 | 7.09 | 7.06 |
| **mPLUG-Owl2 (F)*** | **5.28** | **5.11** | **6.06** |
| Qwen-VL-Chat (B) | 8.23 | 8.19 | 7.99 |
| **Qwen-VL-Chat (F)** | **8.15** | **8.17** | **8.00** |
| LLaVA-CoT (B) | 7.92 | 3.07 | 4.68 |
| **LLaVA-CoT (F)** | **8.04** | **7.95** | **7.76** |

**Chain-Text Probe After Fine-tuning.** Table 6 addresses our central question, whether explicit chain supervision can close the Abstraction Gap.

*Table 6.* Chain-Text Probe: Baseline (B) vs. Fine-tuned (F), averaged across judges. For L2–L3, scores are reported as (Text, Chain). *mPLUG-Owl2 (F): 500 blanks.

| Model | L1 | L2 (T, C) | L3 (T, C) |
|---|---|---|---|
| LLaVA-NeXT (B) | 7.71 | (7.96, 7.83) | (7.64, 7.61) |
| **LLaVA-NeXT (F)** | **7.93** | **(7.88, 7.67)** | **(7.44, 7.28)** |
| MiniGPT-4 (B) | 7.06 | (5.14, 2.09) | (4.91, 2.05) |
| **MiniGPT-4 (F)** | **7.56** | **(6.01, 0.64)** | **(5.78, 0.54)** |
| mPLUG-Owl2 (B) | 6.65 | (5.77, 0.42) | (5.84, 0.75) |
| **mPLUG-Owl2 (F)*** | **5.15** | **(5.20, 0.52)** | **(5.37, 0.82)** |
| Qwen-VL-Chat (B) | 8.01 | (6.86, 3.77) | (6.76, 4.28) |
| **Qwen-VL-Chat (F)** | **7.92** | **(7.04, 5.71)** | **(6.98, 6.47)** |
| LLaVA-CoT (B) | 7.96 | (6.78, 0.97) | (6.81, 1.60) |
| **LLaVA-CoT (F)** | **7.81** | **(6.31, 3.34)** | **(5.70, 3.70)** |

**Key Finding: Fine-tuning does not close the Abstraction Gap for low-baseline models.** The results point to a pronounced asymmetry between text and chain improve-

ment. MiniGPT-4's chain scores *decrease* after fine-tuning (L3: $2.05 \rightarrow 0.54$, $p < 0.001$), the opposite of the intended effect. mPLUG-Owl2 represents no significant chain improvement (L3: $0.75 \rightarrow 0.82$, $p = 0.36$). LLaVA-CoT shows modest improvement but AG remains high (L3: $1.60 \rightarrow 3.70$), insufficient to close the gap. In contrast, Qwen-VL-Chat, which had a moderate baseline AG (0.50), shows significant improvement ($p < 0.001$). Models with some initial structural capability can benefit from chain supervision. LLaVA-NeXT maintains strong performance with no significant change, consistent with its already near-zero AG.

The pattern aligns with the LIMO hypothesis (Ye et al., 2025). Fine-tuning can activate capabilities that exist in a model's representations but cannot create capabilities that are absent. The 45,000 chain-annotated examples ($56\times$ the LIMO threshold of 800 examples) should be sufficient if the limitation were data scarcity. The persistence of high AG values in most models suggests that explicit chain supervision cannot instill structural abstraction capability when it was not developed during earlier training.

## 4.4. Ablation Study: Chain Supervision

To isolate the effect of chain supervision format, we compare models fine-tuned with chains (F+C) versus without chains (F-C) on LLaVA-NeXT and MiniGPT-4.

*Table 7.* Ablation: Chain-Text Probe scores averaged across judges. B = Baseline, F+C = Fine-tuned with chains, F-C = Fine-tuned without chains.

| Model | L1 | L2 (T, C) | L3 (T, C) |
|---|---|---|---|
| LLaVA-NeXT (B) | 7.71 | (7.96, 7.83) | (7.64, 7.61) |
| LLaVA-NeXT (F+C) | 7.93 | (7.88, 7.67) | (7.44, 7.28) |
| LLaVA-NeXT (F-C) | 7.90 | (7.88, 7.69) | (7.46, 7.29) |
| MiniGPT-4 (B) | 7.06 | (5.14, 2.09) | (4.91, 2.05) |
| MiniGPT-4 (F+C) | 7.56 | (6.01, 0.64) | (5.78, 0.54) |
| MiniGPT-4 (F-C) | 7.50 | (5.88, 0.58) | (5.74, 0.53) |

Chain supervision yields no significant improvement in chain generation capability. For MiniGPT-4 L3, chain scores are almost identical under F+C and F-C (0.54 and 0.53, $p = 0.62$). LLaVA-NeXT also shows negligible differences (7.28 vs. 7.29, $p = 0.66$). All comparisons yield $p > 0.05$ (Appendix A.11). The Abstraction Gap cannot be resolved by providing training examples in the target format. The lack of benefit from explicit chain supervision implicates how current VLMs encode (or fail to encode) causal structure during pretraining. Exposure to chain-formatted outputs during fine-tuning does not address the deficit.

## 4.5. Independence of Perceptual and Structural Grounding

The preceding experiments confirm that the Abstraction Gap persists despite extensive fine-tuning on causal chains. A question follows about the nature of the deficit. Does high

AG reflect a general failure of visual grounding, or a specific inability to represent causal structure? If causal abstraction and perceptual grounding share underlying mechanisms, we would expect fine-tuning on causal chains to affect hallucination rates. To test this, we evaluate fine-tuned models on object hallucination benchmarks, MMHal-Bench (Sun et al., 2024) and POPE (Li et al., 2023b).

Table 8 presents hallucination metrics, which remain stable across all fine-tuned models. LLaVA-NeXT maintains identical POPE accuracy (0.86) before and after fine-tuning. MiniGPT-4 exhibits slight improvement on MMHal hallucination rate (0.65 → 0.60) despite its chain generation degrading. Full results across all POPE difficulty levels are in Appendix A.9.

*Table 8.* Hallucination benchmarks: Baseline (B) vs. Fine-tuned (F).

| Model | MMHal Avg | MMHal Hall. | POPE Acc | POPE F1 |
|---|---|---|---|---|
| LLaVA-NeXT (B) | 3.02 | 0.51 | 0.86 | 0.86 |
| **LLaVA-NeXT (F)** | **2.80** | **0.52** | **0.86** | **0.86** |
| MiniGPT-4 (B) | 1.68 | 0.65 | 0.71 | 0.73 |
| **MiniGPT-4 (F)** | **1.88** | **0.60** | **0.70** | **0.75** |
| Qwen-VL-Chat (B) | 2.66 | 0.48 | 0.83 | 0.83 |
| **Qwen-VL-Chat (F)** | **2.64** | **0.43** | **0.82** | **0.82** |
| LLaVA-CoT (B) | 3.05 | 0.41 | 0.85 | 0.84 |
| **LLaVA-CoT (F)** | **2.80** | **0.43** | **0.85** | **0.83** |
| mPLUG-Owl2 (B) | 1.49 | 0.65 | 0.81 | 0.81 |
| **mPLUG-Owl2 (F)** | **1.43** | **0.67** | **0.81** | **0.81** |

The independence has theoretical implications. Hallucination benchmarks test *perceptual grounding*, whether generated content corresponds to objects present in the image. The Abstraction Gap tests *structural grounding*, whether generated causal language reflects underlying relational understanding. The lack of transfer between these capabilities indicates that VLMs maintain separate mechanisms for "what exists" versus "how things relate." LLaVA-RLHF exhibits severe AG (0.85) despite explicit hallucination mitigation training because reducing perceptual hallucination does not improve structural reasoning.

### 4.6. Ablation Study: Comma-Separated Sequences

To examine whether the observed Abstraction Gap is specific to the arrow-chain output format, we evaluate four models using comma-separated sequences (e.g., A, B, C) instead of arrow chains.

*Table 9.* Ablation: AG values when using arrow chains vs. comma-separated sequences, averaged across judges.

| Model | Arrow AG | Comma AG |
|---|---|---|
| LLaVA-NeXT | 0.035 | 0.008 |
| MiniGPT-4 | 0.700 | 0.991 |
| mPLUG-Owl2 | 0.917 | 0.958 |
| Qwen-VL-Chat | 0.501 | 0.400 |

The comma-separated format tests whether output syntax is the primary barrier. Performance worsens for MiniGPT-4

and mPLUG-Owl2, while Qwen-VL-Chat represents only modest improvement and still retains a substantial gap. LLaVA-NeXT remains unaffected. Overall, models that struggle with arrow chains also fail with comma-separated sequences. The performance gap stems not from the specific output format, but from a deeper lack of structural and causal reasoning capabilities in these models.

### 4.7. Ablation Study: Text-First Ordering

We further investigate whether Abstraction Gap values change when models are prompted to generate the textual response first, followed by the causal chain. We evaluate this text-first ordering on the same four models.

*Table 10.* Ablation: AG values when using chain-first ordering vs. text-first ordering, averaged across judges.

| Model | Chain-First AG | Text-First AG |
|---|---|---|
| LLaVA-NeXT | 0.035 | 0.013 |
| MiniGPT-4 | 0.700 | 0.537 |
| mPLUG-Owl2 | 0.917 | 0.742 |
| Qwen-VL-Chat | 0.501 | 0.144 |

All four models show improvement with text-first prompting. Prompt design influences performance. Qwen-VL-Chat exhibits the largest gain (0.501 → 0.144). The model possesses partial structural capability that becomes more accessible through self-scaffolding. However, MiniGPT-4 and mPLUG-Owl2 continue to exhibit substantial gaps (0.537 and 0.742, respectively) even under text-first ordering. These models lack robust structural and causal reasoning abilities, regardless of scaffolding.

### 4.8. Evaluation on Recent Models

The preceding experiments expose a severe Abstraction Gap across most evaluated VLMs. Is this gap an inevitable limitation of current VLM architectures, or can it be overcome? To address this, we evaluate two recent models on the baseline Chain-Text Probe. Qwen3-VL (Bai et al., 2025) is an open-source model. Gemini 2.5 Flash (Comanici et al., 2025) is a closed-source model from Google DeepMind with undisclosed parameter count.

*Table 11.* Recent model evaluation on Chain-Text Probe (Levels 2–3 averaged across judges). Text and Chain scores on 0–10 scale.

| Model | Text | Chain | AG |
|---|---|---|---|
| Qwen3-VL | 7.80 | 5.74 | 0.313 |
| Gemini 2.5 Flash | 8.60 | 8.46 | -0.006 |

The contrast is pronounced. Gemini 2.5 Flash achieves AG of -0.006, effectively eliminating the Abstraction Gap. The model generates causal chains (8.46) at nearly the same quality as its text responses (8.60). Qwen3-VL exhibits a moderate gap (AG = 0.313). The limitation persists in recent open-source models, though Qwen3-VL represents

improvement over most models in our primary evaluation.

The variance across models under identical conditions (from -0.006 to 0.92) confirms that CAGE measures capability differences, not task difficulty. The Abstraction Gap is not an artifact of evaluation design or an inevitable property of VLM architectures. Some models succeed (AG $\approx$ 0) while others fail (AG $>$ 0.7). The gap is closable. Gemini 2.5 Flash uses a sparse mixture-of-experts (MoE) architecture (Comanici et al., 2025), which routes tokens to specialized expert subnetworks and can isolate reasoning capabilities from linguistic processing. In dense architectures, RLHF optimization for fluent text may suppress structural generation because all parameters are affected. MoE could preserve both capabilities by routing them to separate experts. Whether MoE explains the near-zero AG or whether other factors (training data, scale, alignment methods) are responsible remains an open question.

# 5. Discussion

The Abstraction Gap exposes a structural deficit in current VLMs. The ability to generate linguistically plausible causal explanations does not entail the ability to represent causal structure. The finding provides empirical grounding for the plausibility-faithfulness distinction from explainable AI research (Lyu et al., 2024; Agarwal et al., 2024). The gap manifests not only in post-hoc explanations but in the reasoning process itself. For high-stakes applications where causal understanding determines outcomes, this disparity poses concrete risks. Plausible explanations may inspire confidence without reflecting genuine structural comprehension.

Our results complement recent work on chain-of-thought verification (Zhang et al., 2025b), which finds that VLMs achieve up to 68% accuracy when selecting valid causal chains from candidates. The severe AG under generation conditions indicates that verification capability does not transfer to generation capability. VLMs can pattern-match to valid structures when provided as options but cannot construct them independently. The pattern is consistent with the view that current training frameworks prioritize discrimination over construction.

**Implications for VLM Training.** Section 4.2 shows clear differences across the LLaVA family. LLaVA-NeXT achieves near-zero AG ($p < 0.001$ at Level 3). Causal chain generation is possible within current VLMs. However, ablation experiments show that this capability does not stem from specialized diagram training (Liu et al., 2024b). LLaVA-1.5 (Liu et al., 2023a), which shares the same base architecture but lacks diagram training, achieves similarly low AG (0.05 vs. 0.03). The underlying architecture supports structured causal generation. In contrast, LLaVA-

RLHF exhibits high AG (0.84) despite sharing the same architecture. We hypothesize that RLHF optimization for factual accuracy and fluent text suppresses structured outputs.

Section 4.8 reinforces this view. Among recent open-source models, Qwen3-VL shows moderate AG (0.313). Gemini 2.5 Flash eliminates the gap entirely (AG = -0.006). Gemini's MoE architecture routes tokens to specialized subnetworks, isolating reasoning from linguistic processing. In dense architectures, RLHF affects all parameters and may force trade-offs between fluency and structure. AG is not inherent to VLM architectures. Lowering AG depends on architectural choices and alignment methods that preserve structural generation capability.

**Two Forms of Visual Grounding.** Section 4.5 shows that the Abstraction Gap is independent of object hallucination. VLMs maintain separate mechanisms for *perceptual grounding* (what objects exist) and *structural grounding* (how entities relate causally). Reducing hallucination through RLHF does not improve structural reasoning, and training on causal chains does not affect hallucination rates. Interventions targeting one form of grounding should not be expected to transfer to the other.

**Limitations.** CAGE evaluates linear causal chains without branching, cycles, or confounders. Success on simple chains is necessary but not sufficient for complex causal reasoning. Our findings set a lower bound on VLM limitations. The benchmark uses COCO images, so generalization to specialized domains (medical imaging, satellite imagery) remains untested.

The inability to produce causal chains could reflect structural reasoning deficits or format-specific production weaknesses. We tested arrow notation and comma-separated formats, but both share sequential structure. We cannot fully disentangle these explanations without testing diverse representational formats.

Our RLHF suppression hypothesis, while consistent with observed variance, requires further investigation. The MoE hypothesis for Gemini's success is similarly preliminary.

**Future Directions.** First, alignment methods. RLHF appears to suppress structured outputs in dense models. Gemini's MoE architecture may preserve structural capability by routing to specialized experts. Future work should compare MoE and dense architectures systematically and identify alignment techniques that preserve structural generation. Second, evaluation scope. The dual-probe framework generalizes beyond causal reasoning. Applying structure-then-text protocols to spatial, temporal, and compositional reasoning would test whether AG is domain-general. Testing diverse output formats (adjacency lists, graph descriptions) would address the format-specificity limitation noted above.

## Acknowledgments

This research was supported by a standard grant from the U.S. National Science Foundation (NSF DUE 2142558).

## Impact Statement

This paper introduces an evaluation methodology for assessing the faithfulness of causal reasoning in VLMs. Our central finding, that fluent causal language can mask absent structural understanding, has implications for AI safety. In high-stakes applications such as medical diagnosis, autonomous systems, and scientific reasoning, decision-makers may place unwarranted trust in VLM explanations that sound plausible but do not reflect genuine causal comprehension. The Abstraction Gap metric provides a tool for identifying this discrepancy before deployment.

We identify two additional findings with broader relevance. First, the verification-generation asymmetry (68% selection accuracy vs. below 2.5/10 generation) suggests that standard multiple-choice evaluations may overestimate VLM reasoning capabilities. Second, the independence of structural grounding (causal abstraction) and perceptual grounding (object hallucination) indicates that interventions targeting one form of reliability should not be assumed to transfer to the other.

Regarding limitations, our benchmark uses GPT-4o for data generation and evaluation, which may propagate biases present in that system. We mitigate this through multi-annotator validation and human evaluation studies. Findings about specific models reflect capabilities at evaluation time and may not generalize as architectures evolve. The benchmark is released to support reproducible research on trustworthy vision-language systems.

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

# A. Appendix

In this appendix, we provide supplementary material including a detailed review of related work A.1, a description of the evaluation and scoring methodology A.2, the Q&A generation prompt used for CAGE A.3, the evaluation prompts for automated judges in Experiments A and B A.4, detailed result tables with confidence intervals via boostraping A.5, Wilcoxon signed-rank test results ($p$-values) A.6, qualitative examples A.7, a comparison of CAGE with existing benchmarks A.8, an evaluation on standard object hallucination benchmarks A.9, details on the experimental settings A.10, and details on the ablation study A.11.

## A.1. Related Work

We situate CAGE within five areas of related research: (1) causal reasoning benchmarks for VLMs, (2) chain-of-thought reasoning and verification, (3) training methods for visual-linguistic reasoning, (4) object hallucination and visual grounding, and (5) theoretical foundations. The review identifies how existing work addresses different aspects of visual causal reasoning and clarifies where CAGE provides complementary contributions.

### A.1.1. CAUSAL REASONING BENCHMARKS FOR VISION-LANGUAGE MODELS

Causal reasoning is the ability to understand and infer cause-and-effect relationships. It represents a core aspect of intelligence necessary for building AI systems capable of reasoning, planning, and intervention (Pearl & Mackenzie, 2018). While traditional VLM benchmarks (Goyal et al., 2019; Hudson & Manning, 2019) focus on object identification, attribute classification, and spatial reasoning, they are predominantly associational in nature and do not probe a model's ability to infer dynamic cause-and-effect relationships, reason about hypothetical interventions, or understand counterfactual scenarios. Recent work has begun addressing this gap through benchmarks specifically designed for causal and counterfactual reasoning. Each benchmark targets different aspects of causal understanding. We review their contributions and identify where CAGE offers complementary evaluation.

**CausalVLBench.** Komanduri et al. (2025) use Pearl's causal hierarchy to evaluate VLM causal understanding using synthetic images from controlled four-variable physical systems with known ground-truth causal structures. The benchmark employs binary questions about individual causal links and variable predictions for interventions and counterfactuals. The controlled approach allows precise assessment of formal causal inference with unambiguous ground truth. CAGE complements CausalVLBench by testing whether similar causal reasoning capabilities transfer to naturalistic real-world scenes where causal structures are not pre-specified, and models need to abstract relationships from raw visual input.

**InfoCausalQA.** Ka et al. (2025) probe causal reasoning by evaluating VLMs' ability to interpret causal relationships presented within diverse infographics, including charts, diagrams, and flowcharts. The benchmark employs multiple-choice questions spanning associational and interventional reasoning. The approach effectively assesses VLMs' capacity to extract causal implications from structured visual data presentations. CAGE addresses a different challenge. It asks models to generate explicit structural representations of causality from unstructured natural photographic scenes where causal relationships are implicit instead of visually depicted.

**TimeCausality.** Wang et al. (2025) evaluate temporal causal reasoning by focusing on irreversible temporal transformations (e.g., spoilage, rusting, aging) using pairs of "before-and-after" images with binary classification tasks. The approach probes understanding of real-world dynamic changes where temporal evolution is observable across image pairs. CAGE uses static single images and requires models to infer potential dynamics and underlying causal structures from a single snapshot. These approaches test complementary capabilities. TimeCausality assesses the interpretation of observed temporal dynamics, while CAGE assesses inference of latent dynamics from static cues.

**CausalVQA.** Foss et al. (2025) introduce a video-based benchmark probing models' understanding of physical-world causality through question-answer pairs. Models select the correct causal inference from provided alternatives based on observed video dynamics. The benchmark effectively tests causal understanding grounded in temporal visual evidence. CAGE complements CausalVQA by testing causal abstraction from static images and requiring explicit chain generation as output. It assesses whether models can construct (not just select) causal representations.

**CausalVLR.** Liu et al. (2023b) introduce a unified framework and toolbox offering methods for causal relation discovery and inference in visual-linguistic reasoning tasks, including causal inference in VQA, event ordering for video captioning, and causal reasoning for medical report generation. The toolkit advances causal understanding evaluation across diverse applications. CAGE contributes to this ecosystem by providing a benchmark with explicitly annotated causal chains, offering a form of structured supervision for models aiming to achieve deeper causal abstraction, and by introducing the dual-probe methodology for isolating structural reasoning from linguistic fluency.

**CELLO.** Chen et al. (2024a) evaluate VLM causal reasoning across four causal rungs (discovery, association, intervention, counterfactuals) using multiple-choice questions with predefined causal chains derived from Visual Genome scene graphs (Krishna et al., 2017). The benchmark provides a rich evaluation of causal reasoning when chains are available as context. CAGE tests a different capability. It examines whether VLMs can generate lightweight causal chains as structured output without being provided candidate chains. The requirement probes the model's ability to construct and externalize causal representations. The evaluation exposes the Abstraction Gap between text generation fluency and structural abstraction capability.

**MuCR.** Li et al. (2025) evaluate cross-modal causal reasoning using 12,000 synthetic siamese image pairs generated by diffusion models. The benchmark features multiple-choice tasks at image, phrase, and sentence levels. It focuses on visual cue identification for discerning cause-and-effect links. The controlled synthetic approach allows precise manipulation of visual differences. CAGE complements MuCR with real-world COCO images and open-ended chain generation. It tests whether models can abstract causal structure from naturalistic scenes without paired comparisons.

**SPLICE.** Ballout et al. (2025) evaluate event-based reasoning across temporal, causal, spatial, and contextual dimensions using instructional video data. Their findings indicate that VLMs struggle to match human performance and rely more on language priors than visual understanding. The pattern of strong linguistic performance coupled with weaker structural reasoning aligns with the Abstraction Gap we identify in static image causal reasoning. The phenomenon may generalize across visual modalities.

**The Case for Generation.** A methodological distinction runs through these benchmarks. Most employ selection-based evaluation (multiple-choice, binary classification), while CAGE requires generation. The distinction has practical and theoretical significance. In high-stakes deployment scenarios (medical diagnosis, autonomous systems), models need to construct explanations de novo instead of choosing from pre-written options. Selection tasks can be solved through similarity-based discrimination in continuous embedding spaces, while generation requires discrete structural abstraction. The verification-generation asymmetry we observe (68% selection accuracy in Zhang et al. (2025b) vs. <2.5/10 generation scores) indicates that selection performance overestimates genuine causal understanding. Table 12 summarizes methodological differences across benchmarks. These differences reflect distinct evaluation goals, not benchmark quality.

*Table 12.* Methodological comparison of visual causal reasoning benchmarks. Each benchmark targets different aspects of causal understanding. Differences reflect distinct evaluation goals.

| Benchmark | Image Type | Task Format | Chain in Eval | Pearl Levels | Requires Generation |
|---|---|---|---|---|---|
| CausalVLBench | Synthetic | Binary / Prediction | No | 1, 2, 3 | No |
| InfoCausalQA | Infographics | Multiple-choice | No | 1, 2 | No |
| TimeCausality | Real (pairs) | Binary classification | No | 2 | No |
| CausalVQA | Video | Multiple-choice | No | 2 | No |
| CELLO | Real (VG) | Multiple-choice | Provided | 1, 2, 3 | No |
| MuCR | Synthetic (pairs) | Multiple-choice | No | 2 | No |
| SPLICE | Video | Multiple-choice | No | 2 | No |
| **CAGE (ours)** | Real (COCO) | Open-ended + Chain | Generated | 1, 2, 3 | Yes |

### A.1.2. CHAIN-OF-THOUGHT REASONING AND VERIFICATION

Chain-of-Thought (CoT) prompting has advanced LLM reasoning capabilities by guiding models to articulate intermediate steps (Wei et al., 2022). Zero-shot CoT prompting with phrases such as "Let's think step by step" can elicit multi-step

reasoning without few-shot examples (Kojima et al., 2022). Recent research has extended this framework to multimodal settings and assessed CoT reasoning grounded in visual inputs.

**MM-CoT.**    Zhang et al. (2025b) introduce a verification-based benchmark for visual CoT reasoning. It requires models to select valid reasoning chains from adversarial distractors. The triadic A $\to$ B $\to$ C chain format requires discriminating between chains that are: (1) visually grounded and logically coherent, (2) visually inconsistent (hallucinations), or (3) logically incoherent (causal violations). The best-performing model (Claude-Sonnet-4) achieves 68% accuracy on image-based reasoning. The result provides a valuable upper bound on current VLM chain verification capability.

Our work investigates the complementary question of whether models that successfully verify chains can also generate them. The answer seems to be no. Models achieving reasonable verification performance score below 2.5 out of 10 on explicit chain generation in CAGE. The verification-generation asymmetry indicates that selection relies on similarity-based discrimination in continuous embedding spaces, whereas generation demands discrete structural abstraction that current architectures struggle to produce.

**MME-CoT.**    Jiang et al. (2025) benchmark CoT quality, robustness, and efficiency across six domains: math, science, OCR, logic, space-time, and general scenes. The framework assesses reasoning quality through fine-grained evaluation of generated explanations and finds that VLMs struggle to consistently generate high-quality linguistic articulations of reasoning. While MME-CoT evaluates verbose explanatory narratives, CAGE requires concise formal symbolic chains. The difference isolates the challenge of structural abstraction from general linguistic explanation. These approaches are complementary. MME-CoT diagnoses linguistic reasoning quality, while CAGE diagnoses structural abstraction capability.

**Visual Commonsense Reasoning (VCR).**    Zellers et al. (2019) probe complex multi-step logical reasoning requiring models to answer questions and provide rationales grounded in visual scenes. VCR has been influential in showing that visual reasoning extends beyond object identification. However, VCR does not specifically isolate causal inference from broader cognitive abilities, and its multiple-choice format does not test generation capability. CAGE builds on VCR's insight that reasoning quality matters while focusing specifically on causal abstraction through generation-based evaluation.

### A.1.3. TRAINING METHODS FOR VISUAL-LINGUISTIC REASONING

Progress in VLMs is largely attributable to self-supervised and contrastive learning frameworks. Foundational approaches such as SimCLR (Chen et al., 2020) and MoCo (He et al., 2020) show the power of learning representations by contrasting positive and negative pairs. CLIP (Radford et al., 2021) extends this to multimodal settings. It aligns visual and linguistic embeddings through large-scale image-text training. These models set strong baselines for zero-shot generalization but leave gaps in complex reasoning abilities. The gaps motivate research on advanced training methods.

**TripletCLIP.**    Patel et al. (2024) improve compositional reasoning through hard negative mining. They use synthetic vision-language negatives during training. The approach teaches models to distinguish between compositionally challenging concepts to yield gains in attribute-object combination tasks. TripletCLIP advances fine-grained visual discrimination but prioritizes similarity in continuous embedding space instead of discrete structural generation, similar to other contrastive methods.

**CF-VLM.**    Zhang et al. (2026) introduce counterfactual fine-tuning with three specialized loss functions: (1) foundational cross-modal alignment (InfoNCE), (2) counterfactual scenario discrimination (hinge loss enforcing factual pairs > counterfactual pairs), and (3) fine-grained causal discrimination (enforcing factual > minimally-edited counterfactuals). The approach substantially improves compositional reasoning on benchmarks such as ARO (Yuksekgonul et al., 2023) and VLChecklist (Zhao et al., 2023). However, CF-VLM's training objectives optimize similarity-based discrimination. The loss functions enforce that $S(I_{\text{factual}}, T_{\text{factual}}) \gg S(I_{\text{counterfactual}}, T_{\text{factual}})$. The training teaches models to compute better similarity scores. Whether such training transfers to discrete structural generation remains an open question that future work could address by evaluating CF-VLM on CAGE.

**Causal Graphical Models for VLMs.**    Parascandolo et al. (2025) examine how predefined causal graphical models can improve VLM compositional understanding. Their work shows the value of incorporating causal structure for improving VLM performance on downstream tasks. CAGE addresses a complementary question. Instead of providing causal structure

to improve performance, can VLMs generate such structures from visual input? Our finding that most models cannot indicates that current training frameworks do not instill generative causal capabilities.

**Architectural Implications.** These training innovations improve VLM performance on similarity-based tasks, but our experiments show that the Abstraction Gap persists across models with diverse training objectives, including those with explicit CoT supervision (LLaVA-CoT), RLHF-style optimization (LLaVA-RLHF), and counterfactual training data. The persistence points to architectural constraints. Optimizing continuous embeddings for discrimination does not confer the ability to generate discrete symbolic structures. The one model achieving low AG in our evaluation (LLaVA-NeXT) shares the same frozen vision encoder approach as other LLaVA variants. The vision encoder architecture is not the determining factor. We hypothesize that training procedures, especially RLHF optimization for factual accuracy and fluent text, suppress the model's tendency to generate structured, causal-chain-style outputs. The hypothesis points toward alignment methods that preserve a model's ability to generate explicit structural reasoning. Training and alignment procedures should be designed with this distinction in mind.

### A.1.4. OBJECT HALLUCINATION AND VISUAL GROUNDING

Object hallucination, where VLMs generate text describing non-existent objects or details, indicates a lack of reliable visual grounding (Ghosh et al., 2025; Huang et al., 2024; Zhao et al., 2025). Research has attributed hallucination to spurious correlations in training data (Chen et al., 2024b), biases favoring language priors (Zhou et al., 2024; Liu et al., 2024a), generative algorithm errors (Zhou et al., 2024), modality gaps (Jiang et al., 2024a), and deficiencies in visual feature representation (Zhang et al., 2025a).

**Benchmarks.** POPE (Li et al., 2023b) evaluates factual consistency through binary questions about object presence, with adversarial settings testing reliability against misleading context. MMHal-Bench (Sun et al., 2024) provides fine-grained hallucination assessment through detailed response evaluation. HallusionBench (Guan et al., 2024) offers diagnostic tools for disentangling language hallucination from visual illusion and assesses factual grounding across diverse failure modes. AMBER (Wang et al., 2024a) provides additional diagnostic capability spanning generative and discriminative hallucination assessment.

**Mitigation Methods.** Training-based approaches include reflective instruction tuning (Zhang et al., 2025c), instruction tuning with positive and negative samples (Liu et al., 2024a), factually augmented RLHF as implemented in LLaVA-RLHF (Sun et al., 2024), and contrastive learning for cross-modal alignment (Jiang et al., 2024a). Training-free methods include Visual Description Grounded Decoding (Ghosh et al., 2025), attention modification during inference (Huang et al., 2024; Leng et al., 2024), and post-hoc text revision (Zhou et al., 2024).

**Perceptual vs. Structural Grounding.** While hallucination mitigation strategies represent progress in enhancing factual accuracy, they focus on perceptual grounding, whether generated content corresponds to objects present in images. Our work identifies a distinct form of grounding. The Abstraction Gap tests structural grounding, whether causal language reflects relational understanding.

Our experiments show that these capabilities are independent. Fine-tuning on causal chains does not affect hallucination metrics (POPE accuracy, MMHal scores remain stable), and models trained with explicit hallucination mitigation exhibit severe AG. In particular, LLaVA-RLHF (Sun et al., 2024), which uses factually augmented RLHF specifically targeting hallucination reduction, achieves AG of 0.85 in our evaluation. The dissociation indicates that perceptual and structural grounding rely on different mechanisms within VLM architectures, and interventions targeting one form should not be expected to transfer to the other.

### A.1.5. THEORETICAL FOUNDATIONS

**Pearl's Causal Hierarchy.** Our work is grounded in Pearl's causal hierarchy (Pearl, 2009; Pearl & Mackenzie, 2018), which defines three ascending levels: association (observing correlations), intervention (reasoning about effects of actions via the do-operator), and counterfactual (imagining alternative scenarios). The framework provides principled structure for evaluating whether VLMs genuinely understand causality or pattern-match linguistic forms. CAGE operationalizes all three levels, with Level 1 (association) serving as a baseline and Levels 2–3 (intervention, counterfactual) requiring causal chain generation.

**Plausibility vs. Faithfulness.** Our dual-probe methodology builds on the distinction between plausibility and faithfulness in explainable AI (Lyu et al., 2024; Agarwal et al., 2024). Plausible explanations seem logical to humans; faithful explanations accurately represent internal reasoning. Recent work cautions that these properties can diverge (Agarwal et al., 2024), and empirical studies find that LLM self-explanation faithfulness varies by task and model (Madsen et al., 2024). The Abstraction Gap measures this distinction for visual causal reasoning. High text scores with low chain scores indicate plausibility without faithfulness.

**Cognitive Foundations.** Research on human cognition indicates that conceptual structure precedes linguistic expression (Levelt, 1989; Jackendoff, 2002). Speakers construct preverbal representations of relationships before formulating language. By analogy, VLMs with genuine causal understanding should generate structural representations before verbalizing explanations. The Chain-Text Probe tests this hypothesis by requiring structure-then-text output ordering to mirror the cognitive sequence.

**Internal Generative Models.** Cognitive science research indicates that genuine causal understanding relies on internal generative models representing latent causes and supporting structured physical reasoning (Friston, 2010; Battaglia et al., 2013). A defining property of such models is generativity, the capacity to construct structural representations of relationships. CAGE tests this generativity through a minimal sufficiency condition by checking whether models can represent causal relationships using simple arrow notation (e.g., $A \rightarrow B \rightarrow C$). Failure to generate such minimal structures indicates that linguistic plausibility masks absent structural understanding despite fluent textual explanations of the same relationships.

### A.1.6. SUMMARY: CAGE'S COMPLEMENTARY CONTRIBUTIONS

CAGE addresses gaps identified across these research areas while complementing existing benchmarks:

1. **Generation vs. Selection.** Unlike benchmarks using selection-based evaluation (CELLO, MM-CoT, MuCR, Info-CausalQA, CausalVQA), CAGE requires explicit chain generation. The approach exposes the verification-generation asymmetry. Models achieve 68% on chain selection but score below 2.5/10 on chain generation. Selection performance overestimates genuine causal understanding.

2. **Real-World Images.** Unlike synthetic benchmarks (CausalVLBench, MuCR), CAGE uses naturalistic COCO images requiring abstraction from unconstrained visual scenes where causal relationships are implicit, not controlled.

3. **Structural Output.** Unlike text-only evaluation, CAGE requires explicit symbolic chains to help isolate abstraction capability from linguistic fluency. The design allows diagnosis of the plausibility-faithfulness gap in visual causal reasoning.

4. **Dual-Probe Methodology.** The Text-Only and Chain-Text probes provide paired evaluation that single-probe benchmarks cannot offer and quantify the Abstraction Gap as the disparity between linguistic and structural performance.

5. **Training Framework Analysis.** Fine-tuning experiments on 45,000 chain-annotated examples show that explicit chain supervision does not close the Abstraction Gap for most models. Structural abstraction capability cannot be instilled through fine-tuning alone when absent from earlier training.

6. **Grounding Dissociation.** Analysis on hallucination benchmarks (POPE, MMHal-Bench) shows that perceptual and structural grounding are independent capabilities, with LLaVA-RLHF exhibiting severe AG (0.85) despite explicit hallucination mitigation training. The finding has implications for VLM architecture and training design.

### A.2. Detailed Evaluation and Scoring Methodology

We implemented a rigorous, granular scoring system for evaluating VLM responses in both the Text-Only Probe (Experiment A) and the Chain-Text Probe (Experiment B). This system was designed to assess different facets of visual causal reasoning performance, from textual plausibility to the generation of explicit causal structures. The evaluation was guided by dedicated prompts provided to both automated judges (GPT-4o (OpenAI, 2024) and Claude 3.5 Sonnet (Anthropic, 2024)) and human annotators, with strict instructions on scoring criteria and the required output format.

Key aspects of our scoring methodology include:

- **Scoring Scale:** All responses were scored on a 0-10 scale. A score of 0 indicated a missing or entirely irrelevant response. Higher scores reflected increasing accuracy, relevance, logical consistency, and adherence to level-specific requirements, with 10 representing a fully correct, precise, and well-supported answer.

- **Text Response Criteria (Experiments A & B):** Text responses were evaluated on multiple dimensions:
  - *Accuracy:* Factual correctness of statements relative to the provided image and captions.
  - *Relevance:* How directly and completely the response addressed the question asked.
  - *Logical Consistency:* Adherence to common-sense causal reasoning principles and the plausibility of the causal links implied in the text.
  - *Level-Specific Requirements:* Fulfillment of criteria specific to the causal level of the question (L1, L2, L3), such as identifying patterns (L1) or describing plausible outcomes/scenarios (L2/L3). Specific deduction rules were applied for errors appropriate to each level (e.g., factual errors for L1, insufficient causal steps or missing temporal/environmental shifts for L2/L3).

- **Causal Chain Criteria (Experiment B, L2 & L3):** For Level 2 and 3 questions in the Chain-Text Probe (Experiment B), the explicitly generated causal chains were evaluated based on:
  - *Structural Validity:* The output adhered to a proper chain format with arrows connecting causal elements.
  - *Minimum Link Requirements:* The chain included at least one causal link ($\geq 1$) for Level 2 questions and at least two causal links ($\geq 2$) for Level 3 questions to reflect the minimum expected complexity for these causal levels.
  - *Causal Coherence:* The overall chain structure represented a coherent, logical, and plausible causal pathway relevant to the question.

- **Experiment B Specific Scoring Protocol:** For the Chain-Text Probe (Experiment B), each Level 2 and 3 response received two separate scores, one specifically for the generated causal chain based on the criteria above, and another for the text response. This allowed us to differentiate between a model's ability to generate structured causal knowledge and its ability to articulate it linguistically.

- **Chain-Text Consistency (Experiment B):** In addition to scoring text and chain independently, evaluators assessed the consistency and alignment between the generated chain and the accompanying text explanation. Misalignments (e.g., chain contradicting the text, text explaining links not present in the chain) indicated a potential disconnect between structured and linguistic representations. This suggested superficial instead of deeply integrated causal understanding.

- **Strict Output Format:** Judges were strictly instructed to output only the numerical scores in a specific machine-readable format (e.g., '[8][7]...' or '[0][8][6][7]...'), without any additional explanatory text. This ensured structured outputs amenable to quantitative analysis.

The scoring rubrics, detailed criteria, specific deduction rules, and the exact evaluation prompts provided to the automated judges and human annotators are included in Appendix A.4. This rigorous system, combined with the use of multiple independent judges and human validation, enhances the reliability and robustness of our evaluation process.

### A.3. Q&A Generation Prompt

**Role and Task** You are a vision-language model (VLM) tasked with generating 9 questions and corresponding answers about a provided image for a visual causal reasoning task, based on Judea Pearl's Ladder of Causation: Level 1 (Association) identifies observable scene patterns; Level 2 (Intervention) predicts outcomes of a specific change, considering temporal/environmental factors; Level 3 (Counterfactual) reasons about alternate scenarios if a past condition differed, requiring multi-step causal logic.

You will be provided with an image and 5 captions describing the image. Your task is to generate 9 questions (3 per Ladder level) that test a candidate VLM's causal reasoning abilities, along with answers that demonstrate appropriate causal reasoning. Ensure the questions and answers are grounded in the image and captions but push the boundaries of reasoning complexity, particularly for Levels 2 and 3. For Levels 2 and 3, each answer must include a lightweight causal chain (e.g., A $\rightarrow$ B or A $\rightarrow$ B $\rightarrow$ C) followed by a period and a text response that explains the reasoning based on the chain.

**Instructions for Question Generation**

**General Guidelines**

- Generate exactly 9 questions: 3 for Level 1 (Association), 3 for Level 2 (Intervention), and 3 for Level 3 (Counterfactual).

- Base all questions and answers on the provided image and captions, ensuring they are visually relevant but not overly dependent on minute details unavailable in the captions.

- Avoid ambiguity—each question should have a clear focus (e.g., a specific element or condition to reason about).

- Ensure answers are concise (1–2 sentences for the text response) and reflect causal reasoning appropriate to the Ladder level.

**Level-Specific Guidelines**

**Association (Level 1):**   Generate 3 questions that require identifying observable patterns, properties, or relationships in the scene (e.g., colors, positions, weather). Keep these straightforward but tied to the captions. Answers should be text-only, describing the observed features without causal chains.

- Example:
  Question: `What is the color of the grass?`
  Answer: `The color of the grass is green.`

**Intervention (Level 2):**   Generate 3 questions for this image. Each question must:

- Predict the outcome of a direct change (e.g., weather, time, action).

- Focus on temporal shifts (e.g., day to night), environmental dynamics (e.g., rain, wind), or object interactions.

- Require at least one causal link (e.g., `Rain → Wet Surface`), but include additional links if the causal chain requires multiple steps (e.g., 2, 3, or more links).

- Align with Temporal, Spatial, or Interventional categories.

- Answer Format: Provide a causal chain (e.g., `A → B` or more links if needed) followed by a period and a text response (1–2 sentences) explaining the reasoning based on the chain.

- Example:
  Question: `If a sudden fog rolls in, how would the path's visibility change?`
  Answer: `Fog → Reduced Visibility.  The fog obscures the path, making it harder to see.`

**Counterfactual (Level 3):**   Generate 3 questions for this image. Each question must:

- Imagine an alternate scenario (e.g., different time, weather, past event).

- Emphasize temporal shifts (e.g., day to night), environmental dynamics (e.g., snow, drought), and multi-step reasoning.

- Require at least two causal links (e.g., `No Rain → Dry Soil → Cracked Ground`), but include additional links if the causal chain requires multiple steps (e.g., 3, 4, or more links).

- Align with Temporal, Spatial, or Counterfactual categories.

- Answer Format: Provide a causal chain (e.g., `A → B → C` or more links if needed) followed by a period and a text response (1–2 sentences) explaining the reasoning based on the chain.

- Examples:
  Question: `If a storm had hit yesterday, would the beach be empty today?`
  Answer: `Storm → Damaged Area → No Visitors.  The storm's damage would deter visitors, leaving the beach empty.`

**Provided Information** `5 Captions:  {}`

**Task** Generate 9 questions and answers based on the provided image and captions, following the instructions above. Output them in the following format and nothing else:

**Output Format** `[Question 1][Answer 1][Question 2][Answer 2][Question 3][Answer 3][Question 4][Answer 4][Question 5][Answer 5][Question 6][Answer 6][Question 7][Answer 7][Question 8][Answer 8][Question 9][Answer 9]`

## A.4. Evaluation Prompts

### A.4.1. EXPERIMENT A EVALUATION PROMPT

You are a vision-language model (VLM) evaluating a candidate VLM's responses to a visual causal reasoning task based on Judea Pearl's Ladder of Causation: Level 1 (Association) identifies observable scene patterns; Level 2 (Intervention) predicts outcomes of a specific change, considering temporal/environmental factors; Level 3 (Counterfactual) reasons about alternate scenarios if a past condition differed, requiring multi-step causal logic.

You will be provided with an image, 5 captions describing the image, 9 questions about the image (3 per Ladder level, with Level 3 questions emphasizing abstract reasoning), and the candidate VLM's 9 responses. Your task is to evaluate the candidate VLM's responses by assigning each a score from 0 to 10 (0 being the worst, 10 being the best). Use the image, captions, and common-sense reasoning to evaluate accuracy, relevance, and logical consistency.

**Instructions for Evaluation**

**Scoring Criteria:** Assign a score from 0 to 10 based on how well the candidate VLM's response answers the question, considering its accuracy, relevance, and logical consistency with the image, captions, and common-sense reasoning:

- (8–10): Fully correct, addresses question precisely.

- (6–7): Partially correct, addresses question but misses details (e.g., Level 2: "Path wet" without slipperiness).

- (3–5): Incorrect, misinterprets question or contradicts image/captions (e.g., Level 1: "Grass is blue" when green).

- (1–2): Provided but severely flawed (e.g., irrelevant to the question).

- (0): Missing or not provided.

**Deductions:**

- Level 1: Deduct 1–3 points for factual errors (e.g., wrong color, position).

- Level 2: Deduct 2–4 points if the response omits environmental/temporal shifts (e.g., no mention of rain's effect).

- Level 3: Deduct 3–5 points if the response omits multi-step reasoning/shifts.

- All Levels: Deduct 1–2 points for overgeneralization (e.g., vague "things change") or ignoring image/captions.

**Use the Image and Captions:** Ensure responses are grounded in the image and captions for Level 1 questions. For Levels 2 and 3, expect reasonable speculation about changes (e.g., weather, time) that align with common-sense dynamics.

**Evaluation Guidelines:**

- Level 1 (Association): Check if the response accurately describes observable features (e.g., colors, positions) based on the image and captions. Deduct points for factual errors.

- Level 2 (Intervention): Verify that the response predicts a plausible outcome of the change (e.g., 'If rain starts, the path becomes wet'). Deduct points if the response lacks causal reasoning or fails to simulate environmental/temporal shifts.

- Level 3 (Counterfactual): Confirm that the response reconstructs a plausible alternate scenario (e.g., 'If it were midnight, the stars would be visible'). Deduct points if the response lacks multi-step reasoning or fails to simulate environmental/temporal shifts.

**Output Format Instructions** Output your evaluation in exactly this format, with no additional text, line breaks, spaces, explanations, or deviations:

- [Rating for Response 1][Rating for Response 2][Rating for Response 3][Rating for Response 4][Rating for Response 5][Rating for Response 6][Rating for Response 7][Rating for Response 8][Rating for Response 9]

- Each [Rating] is a number from 0 to 10 in square brackets (e.g., [7]).

- Do not include explanations, extra spaces, line breaks, headings, or any text outside this format.

Example Output: [8][7][9][8][9][6][9][5][8]

**Provided Information** 5 Captions: {}

9 pairs of Questions and Candidate VLM's Responses: {}

**Task** Evaluate the candidate VLM's responses to the 9 questions using the provided image, captions, and instructions above. Output your evaluation in the following format and nothing else:

**Output Format** [Rating for Response 1][Rating for Response 2][Rating for Response 3][Rating for Response 4][Rating for Response 5][Rating for Response 6][Rating for Response 7][Rating for Response 8][Rating for Response 9]

A.4.2. EXPERIMENT B EVALUATION PROMPT

You are a vision-language model (VLM) evaluating a candidate VLM's responses to a visual causal reasoning task based on Judea Pearl's Ladder of Causation: Level 1 (Association) identifies observable scene patterns; Level 2 (Intervention) predicts outcomes of a specific change, considering temporal/environmental factors; Level 3 (Counterfactual) reasons about alternate scenarios if a past condition differed, requiring multi-step causal logic.

You will be provided with an image, 5 captions describing the image, 9 questions about the image (3 per Ladder level, with Level 3 questions emphasizing abstract reasoning), and the candidate VLM's 9 responses. For Levels 2 and 3, each response is expected to include a lightweight causal chain (e.g., A $\rightarrow$ B or A $\rightarrow$ B $\rightarrow$ C) to explain the reasoning. Your task is to evaluate each response by assigning two separate scores from 0 to 10: one for the causal chain (if applicable) and one for the text response, using the image, captions, common-sense reasoning, and, for Levels 2 and 3, the causal chain to judge accuracy, relevance, and logical consistency. No ground truth answers are provided, but you must rely on your understanding of the scene and causal principles to make informed judgments.

**Instructions for Evaluation**

**Scoring Criteria:** Assign a score from 0 to 10 based on how well the candidate VLM's response answers the question, considering its accuracy, relevance, and logical consistency with the image, captions, and common-sense reasoning. For Levels 2 and 3, also evaluate the causal chain and its alignment with the text.

- **Causal Chain (Levels 2 and 3 only):**
  - (8–10): Fully correct, relevant, with required links (Level 2: $\geq$1, e.g., Rain $\rightarrow$ Wet Path; Level 3: $\geq$2, e.g., Midnight $\rightarrow$ Dark Sky $\rightarrow$ Stars).
  - (6–7): Partially correct, misses minor links or has small inaccuracies (e.g., Level 3: Midnight $\rightarrow$ Stars, omitting Dark Sky).
  - (3–5): Incorrect, with illogical or irrelevant links (e.g., Rain $\rightarrow$ Sun).
  - (1–2): Provided but severely flawed (e.g., wrong direction, no causal logic).

   – (0): Missing or not provided.

- **Text Response (All Levels):**

   – (8–10): Fully correct, addresses question precisely, aligns with chain for Levels 2 and 3 (e.g., Level 3: "Stars visible" with Midnight $\rightarrow$ Dark Sky $\rightarrow$ Stars).
   – (6–7): Partially correct, addresses question but misses details or minor causal steps (e.g., Level 2: "Path wet" without slipperiness).
   – (3–5): Incorrect, misinterprets question or contradicts image/captions (e.g., Level 1: "Grass is blue" when green).
   – (1–2): Provided but severely flawed (e.g., irrelevant to question).
   – (0): Missing or not provided.

**Deductions:**

- Level 1: Deduct 1–3 points for factual errors (e.g., wrong color, position).

- Level 2: Deduct 2–4 points if chain lacks $\geq 1$ link or text omits environmental/temporal shifts (e.g., no mention of rain's effect).

- Level 3: Deduct 3–5 points if chain lacks $\geq 2$ links (e.g., Snow $\rightarrow$ Ground White, missing Cold $\rightarrow$ Icicles) or text omits multi-step reasoning/shifts.

- All Levels: Deduct 1–2 points for overgeneralization (e.g., vague "things change") or ignoring image/captions.

**Use the Image and Captions:** Ensure responses are grounded in the image and captions for Level 1 questions. For Levels 2 and 3, expect reasonable speculation about changes (e.g., weather, time) that align with common-sense dynamics.

**Evaluation Guidelines:**

- Level 1 (Association): Score text for accuracy against image/captions (e.g., "What color is the grass?" $\rightarrow$ "Green" if green). No chains are expected. Set chain score to [0].

- Level 2 (Intervention): Score text for plausible outcome (e.g., "If rain starts, path becomes wet") and chain for $\geq 1$ logical link (e.g., Rain $\rightarrow$ Wet Path). Ensure text and chain align (e.g., text mentions wetness, chain includes it). Check for temporal/environmental shifts (e.g., weather change).

- Level 3 (Counterfactual): Score text for plausible alternate scenario (e.g., "If midnight, stars visible") and chain for $\geq 2$ logical links (e.g., Midnight $\rightarrow$ Dark Sky $\rightarrow$ Stars). Ensure text and chain align and reflect multi-step reasoning (e.g., temporal shift to night). Deduct heavily for missing shifts or single-link chains.

**Output Format Instructions** Output your evaluation in exactly this format, with no additional text, line breaks, spaces, explanations, or deviations:

- [Chain Rating for Response 1][Text Rating for Response 1][Chain Rating for Response 2][Text Rating for Response 2][Chain Rating for Response 3][Text Rating for Response 3][Chain Rating for Response 4][Text Rating for Response 4][Chain Rating for Response 5][Text Rating for Response 5][Chain Rating for Response 6][Text Rating for Response 6][Chain Rating for Response 7][Text Rating for Response 7][Chain Rating for Response 8][Text Rating for Response 8][Chain Rating for Response 9][Text Rating for Response 9]

- Each [Chain Rating] and [Text Rating] is a number from 0 to 10 in square brackets (e.g., [7]).

- For Level 1, use [0] for the chain score.

- Do not include explanations, extra spaces, line breaks, headings, or any text outside this format.

Example Output: `[0][8][0][7][0][9][6][8][8] [9][7][6][9][9][5][5][8][8]`

**Provided Information** `5 Captions: {}`

`9 pairs of Questions and Candidate VLM's Responses: {}`

**Task** Evaluate the candidate VLM's responses to the 9 questions using the provided image, captions, and instructions above. Assign separate scores for the causal chain and text response for each, and output your evaluation in the following format and nothing else:

**Output Format** `[Chain Rating for Response 1][Text Rating for Response 1][Chain Rating for Response 2][Text Rating for Response 2][Chain Rating for Response 3][Text Rating for Response 3][Chain Rating for Response 4][Text Rating for Response 4][Chain Rating for Response 5][Text Rating for Response 5][Chain Rating for Response 6][Text Rating for Response 6][Chain Rating for Response 7][Text Rating for Response 7][Chain Rating for Response 8][Text Rating for Response 8][Chain Rating for Response 9][Text Rating for Response 9]`

### A.5. Experimental Results with Confidence Intervals

Tables 13–18 provide the results of the experiments conducted in the main paper with the 95% confidence intervals recorded and per-judge breakdown.

*Table 13.* Baseline Average Evaluation Scores (0–10) for Text-Only Probe (Experiment A) with 95% confidence intervals on 500 COCO images, evaluated by GPT-4o and Claude 3.5 Sonnet. Scores are averaged over 500 images and 3 questions per difficulty level. Blank responses are scored as 0. *mPLUG-Owl2 (249 blanks) and LLaVA-RLHF (2467 blanks) yield high numbers of blank responses.

| | GPT-4o Eval | | | Claude 3.5 Sonnet Eval | | |
|---|---|---|---|---|---|---|
| **VLM** | **Avg L1** | **Avg L2** | **Avg L3** | **Avg L1** | **Avg L2** | **Avg L3** |
| CogVLM2 | 8.17 (95% CI 8.06-8.28) | 8.42 (95% CI 8.35-8.48) | 8.41 (95% CI 8.35-8.48) | 8.58 (95% CI 8.54-8.68) | 8.09 (95% CI 8.06-8.16) | 7.80 (95% CI 7.77-7.88) |
| LLaVA-NeXT | 7.85 (95% CI 7.72-7.97) | 8.29 (95% CI 8.21-8.37) | 8.29 (95% CI 8.22-8.37) | 8.05 (95% CI 7.98-8.16) | 7.87 (95% CI 7.83-7.96) | 7.58 (95% CI 7.53-7.67) |
| MiniGPT-4 | 6.75 (95% CI 6.61-6.89) | 7.24 (95% CI 7.13-7.34) | 6.73 (95% CI 6.61-6.85) | 7.44 (95% CI 7.33-7.54) | 7.04 (95% CI 6.95-7.12) | 6.63 (95% CI 6.54-6.73) |
| mPLUG-Owl2* | 6.50 (95% CI 6.33-6.67) | 7.22 (95% CI 7.09-7.34) | 7.31 (95% CI 7.20-7.42) | 7.10 (95% CI 6.96-7.24) | 6.95 (95% CI 6.88-7.10) | 6.80 (95% CI 6.72-6.92) |
| Qwen-VL-Chat | 8.02 (95% CI 7.89-8.14) | 8.30 (95% CI 8.23-8.37) | 8.28 (95% CI 8.21-8.35) | 8.43 (95% CI 8.36-8.53) | 8.07 (95% CI 8.05-8.16) | 7.70 (95% CI 7.66-7.79) |
| InternVL2 | 7.88 (95% CI 7.75-8.00) | 7.86 (95% CI 7.76-7.94) | 7.46 (95% CI 7.36-7.56) | 8.06 (95% CI 7.96-8.16) | 7.27 (95% CI 7.20-7.35) | 6.77 (95% CI 6.69-6.88) |
| LLaVA-RLHF* | 4.34 (95% CI 4.17-4.50) | 7.06 (95% CI 6.95-7.16) | 7.41 (95% CI 7.31-7.50) | 5.89 (95% CI 5.80-6.09) | 6.46 (95% CI 6.40-6.59) | 6.43 (95% CI 6.36-6.55) |
| LLaVA-CoT | 7.76 (95% CI 7.63-7.88) | 2.93 (95% CI 2.74-3.11) | 4.65 (95% CI 4.45-4.86) | 8.07 (95% CI 7.98-8.18) | 3.21 (95% CI 2.97-3.32) | 4.70 (95% CI 4.51-4.87) |

### A.6. Wilcoxon Signed-Rank Test Results

We apply the Wilcoxon signed-rank test to evaluate three core hypotheses regarding the Abstraction Gap (AG) and the effects of fine-tuning and chain supervision:

- **H1**: The AG is statistically significant (text outperforms chain generation).

- **H2**: Fine-tuning significantly improves performance.

- **H3**: Chain supervision during fine-tuning significantly enhances chain generation.

*Table 14.* Baseline Average Evaluation Scores (0–10) for Chain-Text Probe (Experiment B) with 95% confidence intervals on 500 COCO images, judged by GPT-4o and Claude 3.5 Sonnet. Scores are averaged over 500 images and 3 questions per level. For L2–L3, results appear as (Text Score (T), Chain Score (C)). Blank responses score 0. *CogVLM2 (532) and LLaVA-RLHF (2467) yield many blank responses. Chain scores in **bold**.

| VLM | GPT-4o Eval | | | Claude 3.5 Sonnet Eval | | |
| --- | --- | --- | --- | --- | --- | --- |
| | Avg L1 T | Avg L2 (T, C) | Avg L3 (T, C) | Avg L1 T | Avg L2 (T, C) | Avg L3 (T, C) |
| CogVLM2* | 8.22 (95% CI 8.15-8.33) | (6.07, **0.09**) (95% CI 5.91-6.18, 95% CI 0.06-0.14) | (5.80, **0.98**) (95% CI 5.67-5.96, 95% CI 0.85-1.07) | 8.07 (95% CI 8.01-8.15) | (5.04, **0.99**) (95% CI 4.91-5.15, 95% CI 0.87-1.11) | (4.80, **1.76**) (95% CI 4.66-4.92, 95% CI 1.58-1.88) |
| LLaVA-NeXT | 8.04 (95% CI 7.94-8.15) | (8.25, **7.95**) (95% CI 8.19-8.31, 95% CI 7.87-8.02) | (7.85, **7.42**) (95% CI 7.79-7.92, 95% CI 7.35-7.51) | 7.37 (95% CI 7.29-7.46) | (7.66, **7.70**) (95% CI 7.62-7.70, 95% CI 7.66-7.74) | (7.42, **7.79**) (95% CI 7.38-7.48, 95% CI 7.75-7.84) |
| MiniGPT-4 | 7.04 (95% CI 6.93-7.17) | (5.22, **1.57**) (95% CI 5.09-5.37, 95% CI 1.46-1.75) | (5.01, **1.52**) (95% CI 4.89-5.16, 95% CI 1.40-1.67) | 7.07 (95% CI 6.97-7.16) | (5.06, **2.60**) (95% CI 4.97-5.20, 95% CI 2.43-2.76) | (4.80, **2.57**) (95% CI 4.70-4.93, 95% CI 2.44-2.78) |
| mPLUG-Owl2 | 6.70 (95% CI 6.53-6.85) | (6.16, **0.22**) (95% CI 6.04-6.25, 95% CI 0.16-0.28) | (6.15, **0.49**) (95% CI 6.12-6.30, 95% CI 0.42-0.59) | 6.59 (95% CI 6.45-6.72) | (5.38, **0.62**) (95% CI 5.30-5.45, 95% CI 0.50-0.70) | (5.53, **1.00**) (95% CI 5.45-5.63, 95% CI 0.87-1.12) |
| Qwen-VL-Chat | 8.13 (95% CI 8.05-8.24) | (7.07, **3.40**) (95% CI 7.01-7.15, 95% CI 3.19-3.58) | (7.00, **4.21**) (95% CI 6.95-7.10, 95% CI 4.06-4.41) | 7.89 (95% CI 7.83-7.97) | (6.64, **4.14**) (95% CI 6.57-6.70, 95% CI 3.92-4.28) | (6.51, **4.34**) (95% CI 6.45-6.57, 95% CI 4.14-4.51) |
| InternVL2 | 8.04 (95% CI 7.95-8.15) | (7.98, **3.92**) (95% CI 7.93-8.05, 95% CI 3.69-4.13) | (7.63, **2.42**) (95% CI 7.59-7.71, 95% CI 2.25-2.62) | 7.49 (95% CI 7.39-7.57) | (6.92, **3.87**) (95% CI 6.85-6.98, 95% CI 3.68-4.09) | (6.76, **2.60**) (95% CI 6.71-6.83, 95% CI 2.42-2.79) |
| LLaVA-RLHF* | 5.06 (95% CI 4.92-5.20) | (1.23, **0.51**) (95% CI 1.10-1.37, 95% CI 0.41-0.60) | (1.92, **1.11**) (95% CI 1.77-2.10, 95% CI 0.98-1.24) | 5.15 (95% CI 5.04-5.30) | (1.43, **1.11**) (95% CI 1.28-1.57, 95% CI 0.97-1.24) | (1.85, **1.52**) (95% CI 1.68-1.99, 95% CI 1.34-1.65) |
| LLaVA-CoT | 8.06 (95% CI 7.96-8.16) | (7.26, **0.77**) (95% CI 7.21-7.32, 95% CI 0.66-0.90) | (7.16, **1.34**) (95% CI 7.10-7.23, 95% CI 1.22-1.51) | 7.86 (95% CI 7.79-7.94) | (6.30, **1.17**) (95% CI 6.26-6.37, 95% CI 1.04-1.32) | (6.45, **1.86**) (95% CI 6.40-6.52, 95% CI 1.70-2.04) |

*Table 15.* Human Evaluation Scores (0–10) for LLaVA-NeXT 13B and MiniGPT-4 13B on 125 COCO images with 95% confidence intervals. Scores average over 3 questions per causal level. For Experiment B L2–L3, results appear as (Text Score (T), Chain Score (C)); chain scores are bolded.

| VLM | Exp A (T) | | | Exp B (T, C) | | |
| --- | --- | --- | --- | --- | --- | --- |
| | L1 | L2 | L3 | L1 T | L2 (T, C) | L3 (T, C) |
| LLaVA-NeXT | 8.94 (95% CI 8.73-9.14) | 9.02 (95% CI 8.88-9.15) | 8.80 (95% CI 8.64-8.95) | 9.07 (95% CI 8.88-9.25) | (8.82, **7.98**) (95% CI 8.68-8.95, 95% CI 7.80-8.17) | (8.49, **7.58**) (95% CI 8.34-8.62, 95% CI 7.37-7.77) |
| MiniGPT-4 | 7.91 (95% CI 7.62-8.19) | 8.21 (95% CI 8.01-8.42) | 7.63 (95% CI 7.40-7.84) | 7.90 (95% CI 7.61-8.18) | (6.52, **1.15**) (95% CI 6.22-6.82, 95% CI 0.93-1.38) | (6.02, **0.95**) (95% CI 5.72-6.33, 95% CI 0.76-1.17) |

## H1: ABSTRACTION GAP IS SIGNIFICANT

Table 19 reports *p*-values from paired comparisons of text-only scores (Experiment A) against chain scores (Experiment B) at levels L2 and L3. Across all VLMs and both judges (GPT-4o and Claude 3.5 Sonnet), $p < 0.0001$. This provides overwhelming evidence that text generation systematically outperforms chain generation. This confirms that AG is not due to random variation.

## H2: MIXED EFFECTS OF FINE-TUNING

Table 20 compares baseline and fine-tuned models across conditions. Fine-tuning yields highly heterogeneous outcomes:

- LLaVA-NeXT: Text gains in Experiment A are marginal and judge-dependent (GPT-4o: $p = 0.0971$; Claude 3.5: $p = 0.0001$). In Experiment B, chain performance significantly degrades ($p < 0.001$ for L2/L3 under both judges).

- LLaVA-CoT: No significant text improvement in Experiment A under GPT-4o ($p = 0.2295$), but strong gains in

*Table 16.* Baseline (B) vs. Fine-tuned (F) Average Evaluation Scores (0–10) for Text-Only Probe (Experiment A) with 95% confidence intervals, evaluated by GPT-4o and Claude 3.5 Sonnet. Higher scores are generally better. Bold indicates fine-tuned models. Shaded rows group models. *Baseline mPLUG-Owl2 (249 blanks) and Fine-tuned mPLUG-Owl2 (1095 blanks) yield high numbers of blank responses.

| VLM | GPT-4o Eval | | | Claude 3.5 Sonnet Eval | | |
|---|---|---|---|---|---|---|
| | Avg L1 | Avg L2 | Avg L3 | Avg L1 | Avg L2 | Avg L3 |
| LLaVA-NeXT (B) | 7.85 (95% CI 7.72-7.97) | 8.29 (95% CI 8.21-8.37) | 8.29 (95% CI 8.22-8.37) | 8.05 (95% CI 7.98-8.16) | 7.87 (95% CI 7.83-7.96) | 7.58 (95% CI 7.53-7.67) |
| **LLaVA-NeXT (F)** | **7.99 (95% CI 7.87-8.11)** | **8.50 (95% CI 8.44-8.56)** | **8.40 (95% CI 8.33-8.47)** | **8.28 (95% CI 8.21-8.38)** | **8.11 (95% CI 8.07-8.18)** | **7.76 (95% CI 7.71-7.84)** |
| MiniGPT-4 (B) | 6.75 (95% CI 6.61-6.89) | 7.24 (95% CI 7.13-7.34) | 6.73 (95% CI 6.61-6.85) | 7.44 (95% CI 7.33-7.54) | 7.04 (95% CI 6.95-7.12) | 6.63 (95% CI 6.54-6.73) |
| **MiniGPT-4 (F)** | **7.31 (95% CI 7.18-7.44)** | **8.19 (95% CI 8.12-8.27)** | **8.11 (95% CI 8.04-8.18)** | **7.87 (95% CI 7.77-7.97)** | **7.70 (95% CI 7.65-7.77)** | **7.45 (95% CI 7.39-7.51)** |
| mPLUG-Owl2 (B)* | 6.50 (95% CI 6.33-6.67) | 7.22 (95% CI 7.09-7.34) | 7.31 (95% CI 7.20-7.42) | 7.10 (95% CI 6.96-7.24) | 6.95 (95% CI 6.88-7.10) | 6.80 (95% CI 6.72-6.92) |
| **mPLUG-Owl2 (F)*** | **4.86 (95% CI 4.66-5.07)** | **5.20 (95% CI 5.01-5.38)** | **6.34 (95% CI 6.20-6.49)** | **5.70 (95% CI 5.48-5.85)** | **5.01 (95% CI 4.89-5.23)** | **5.78 (95% CI 5.66-5.93)** |
| Qwen-VL-Chat (B) | 8.02 (95% CI 7.89-8.14) | 8.30 (95% CI 8.23-8.37) | 8.28 (95% CI 8.21-8.35) | 8.43 (95% CI 8.36-8.53) | 8.07 (95% CI 8.05-8.16) | 7.70 (95% CI 7.66-7.79) |
| **Qwen-VL-Chat (F)** | **8.04 (95% CI 7.92-8.16)** | **8.47 (95% CI 8.39-8.53)** | **8.35 (95% CI 8.28-8.41)** | **8.25 (95% CI 8.18-8.35)** | **7.87 (95% CI 7.84-7.94)** | **7.65 (95% CI 7.59-7.70)** |
| LLaVA-CoT (B) | 7.76 (95% CI 7.63-7.88) | 2.93 (95% CI 2.74-3.11) | 4.65 (95% CI 4.45-4.86) | 8.07 (95% CI 7.98-8.18) | 3.21 (95% CI 2.97-3.32) | 4.70 (95% CI 4.51-4.87) |
| **LLaVA-CoT (F)** | **7.82 (95% CI 7.70-7.94)** | **8.11 (95% CI 8.00-8.22)** | **8.05 (95% CI 7.94-8.16)** | **8.26 (95% CI 8.20-8.37)** | **7.79 (95% CI 7.72-7.91)** | **7.46 (95% CI 7.39-7.58)** |

Experiment B for both text and chain ($p < 0.001$).

- MiniGPT-4: Text scores improve dramatically ($p < 0.001$), but chain generation collapses catastrophically ($p < 0.001$).

- Qwen-VL-Chat: Text changes in Experiment A are significant but directionally inconsistent across judges; chain and text in Experiment B improve ($p < 0.001$).

- mPLUG-Owl2: Text performance worsens significantly ($p < 0.001$); chain scores show minor mean gains but remain non-significant ($p > 0.14$).

These results indicate that standard fine-tuning does not reliably support structured causal chain generation and may even exacerbate the AG in some models.

H3: CHAIN SUPERVISION HAS NO SIGNIFICANT IMPACT

Table 21 compares fine-tuned models trained *with* vs. *without* chain supervision. All $p$-values exceed 0.05 (ranging from 0.0887 to 0.9153), which indicates no statistically significant difference in chain generation performance. This suggests that explicit chain-structured training data is not sufficient to overcome the AG and points to deeper architectural or representational limitations.

### A.7. Qualitative Examples

This subsection presents three concrete examples (Figures 3–5) that illustrate how LLaVA-NeXT and MiniGPT-4 respond to causal questions on images from the CAGE benchmark on the Chain-Text Probe (Experiment B). Each case shows the input image, the causal question, the full response from both models, and the scores assigned by GPT-4o for the causal chain and the textual answer. These examples allow a comparison of the models' ability to generate structured causal reasoning and show the score ranges GPT-4o assigns.

### A.8. Comparison to Existing Benchmarks

CAGE contributes uniquely to the landscape of reasoning benchmarks. Table 22 compares CAGE with relevant existing datasets across important dimensions. Unlike benchmarks that implicitly address causal aspects or focus on multiple-choice

*Table 17.* Baseline (B) vs. Fine-tuned (F) Average Evaluation Scores (0–10) for Chain-Text Probe (Experiment B) with confidence intervals, evaluated by GPT-4o and Claude 3.5 Sonnet. For L2 and L3, scores are (Text (T), Chain (C)). Higher scores are better. Bold indicates fine-tuned models and chain scores. Shaded rows group models. *Fine-tuned mPLUG-Owl2 (500 blanks) yields a high number of blank responses.

| VLM | GPT-4o Eval | | | Claude 3.5 Sonnet Eval | | |
|---|---|---|---|---|---|---|
| | Avg L1 T | Avg L2 (T, C) | Avg L3 (T, C) | Avg L1 T | Avg L2 (T, C) | Avg L3 (T, C) |
| LLaVA-NeXT (B) | 8.04 (95% CI 7.94-8.15) | (8.25, **7.95**) (95% CI 8.19-8.31, 95% CI 7.87-8.02) | (7.85, **7.42**) (95% CI 7.79-7.92, 95% CI 7.35-7.51) | 7.37 (95% CI 7.29-7.46) | (7.66, **7.70**) (95% CI 7.62-7.70, 95% CI 7.66-7.74) | (7.42, **7.79**) (95% CI 7.38-7.48, 95% CI 7.75-7.84) |
| **LLaVA-NeXT (F)** | **8.17** **(95% CI 8.07-8.27)** | **(8.16, 7.75)** **(95% CI 8.09-8.22, 95% CI 7.67-7.84)** | **(7.59, 6.96)** **(95% CI 7.53-7.67, 95% CI 6.88-7.06)** | **7.68** **(95% CI 7.61-7.77)** | **(7.60, 7.58)** **(95% CI 7.58-7.66, 95% CI 7.55-7.63)** | **(7.29, 7.59)** **(95% CI 7.26-7.36, 95% CI 7.55-7.65)** |
| MiniGPT-4 (B) | 7.04 (95% CI 6.93-7.17) | (5.22, **1.57**) (95% CI 5.09-5.37, 95% CI 1.46-1.75) | (5.01, **1.52**) (95% CI 4.89-5.16, 95% CI 1.40-1.67) | 7.07 (95% CI 6.97-7.16) | (5.06, **2.60**) (95% CI 4.97-5.20, 95% CI 2.43-2.76) | (4.80, **2.57**) (95% CI 4.70-4.93, 95% CI 2.44-2.78) |
| **MiniGPT4 (F)** | **7.63** **(95% CI 7.50-7.73)** | **(6.32, 0.34)** **(95% CI 6.23-6.39, 95% CI 0.25-0.38)** | **(5.98, 0.18)** **(95% CI 5.93-6.09, 95% CI 0.09-0.17)** | **7.49** **(95% CI 7.40-7.57)** | **(5.70, 0.94)** **(95% CI 5.62-5.76, 95% CI 0.81-1.05)** | **(5.58, 0.89)** **(95% CI 5.51-5.65, 95% CI 0.74-0.98)** |
| mPLUG-Owl2 (B) | 6.70 (95% CI 6.53-6.85) | (6.16, **0.22**) (95% CI 6.04-6.25, 95% CI 0.16-0.28) | (6.15, **0.49**) (95% CI 6.12-6.30, 95% CI 0.42-0.59) | 6.59 (95% CI 6.45-6.72) | (5.38, **0.62**) (95% CI 5.30-5.45, 95% CI 0.50-0.70) | (5.53, **1.00**) (95% CI 5.45-5.63, 95% CI 0.87-1.12) |
| **mPLUG-Owl2 (F)*** | **5.04** **(95% CI 4.80-5.19)** | **(5.50, 0.30)** **(95% CI 5.36-5.62, 95% CI 0.23-0.37)** | **(5.65, 0.57)** **(95% CI 5.52-5.75, 95% CI 0.49-0.67)** | **5.26** **(95% CI 5.04-5.38)** | **(4.90, 0.74)** **(95% CI 4.80-4.99, 95% CI 0.63-0.84)** | **(5.09, 1.06)** **(95% CI 4.99-5.19, 95% CI 0.93-1.19)** |
| Qwen-VL-Chat (B) | 8.13 (95% CI 8.05-8.24) | (7.07, **3.40**) (95% CI 7.01-7.15, 95% CI 3.19-3.58) | (7.00, **4.21**) (95% CI 6.95-7.10, 95% CI 4.06-4.41) | 7.89 (95% CI 7.83-7.97) | (6.64, **4.14**) (95% CI 6.57-6.70, 95% CI 3.92-4.28) | (6.51, **4.34**) (95% CI 6.45-6.57, 95% CI 4.14-4.51) |
| **QwenVL-Chat (F)** | **8.22** **(95% CI 8.14-8.33)** | **(7.43, 5.77)** **(95% CI 7.35-7.52, 95% CI 5.60-5.95)** | **(7.31, 6.51)** **(95% CI 7.24-7.38, 95% CI 6.38-6.63)** | **7.62** **(95% CI 7.56-7.71)** | **(6.64, 5.64)** **(95% CI 6.58-6.71, 95% CI 5.44-5.76)** | **(6.65, 6.43)** **(95% CI 6.60-6.72, 95% CI 6.34-6.59)** |
| LLaVA-CoT (B) | 8.06 (95% CI 7.96-8.16) | (7.26, **0.77**) (95% CI 7.21-7.32, 95% CI 0.66-0.90) | (7.16, **1.34**) (95% CI 7.10-7.23, 95% CI 1.22-1.51) | 7.86 (95% CI 7.79-7.94) | (6.30, **1.17**) (95% CI 6.26-6.37, 95% CI 1.04-1.32) | (6.45, **1.86**) (95% CI 6.40-6.52, 95% CI 1.70-2.04) |
| **LLaVA-CoT (F)** | **7.92** **(95% CI 7.83-8.03)** | **(6.74, 2.99)** **(95% CI 6.62-6.90, 95% CI 2.84-3.24)** | **(6.12, 3.67)** **(95% CI 5.99-6.30, 95% CI 3.51-3.90)** | **7.70** **(95% CI 7.65-7.80)** | **(5.88, 3.69)** **(95% CI 5.76-6.01, 95% CI 3.52-3.90)** | **(5.29, 3.73)** **(95% CI 5.15-5.44, 95% CI 3.53-3.90)** |

*Table 18.* Ablation Study: Experiment B (Chain-Text Probe) Scores (0–10) with 95% confidence intervals, evaluated by GPT-4o and Claude 3.5 Sonnet. For L2 and L3, scores are (Text (T), Chain (C)). **Abbreviations:** B = Baseline; F+C = Fine-tuned with chain supervision; F-C = Fine-tuned without chain supervision. Shaded rows group models.

| VLM | GPT-4o Eval | | | Claude 3.5 Sonnet Eval | | |
|---|---|---|---|---|---|---|
| | Avg L1 T | Avg L2 (T, C) | Avg L3 (T, C) | Avg L1 T | Avg L2 (T, C) | Avg L3 (T, C) |
| LLaVA-NeXT (B) | 8.04 (95% CI 7.94-8.15) | (8.25, 7.95) (95% CI 8.19-8.31, 95% CI 7.87-8.02) | (7.85, 7.42) (95% CI 7.79-7.92, 95% CI 7.35-7.51) | 7.37 (95% CI 7.29-7.46) | (7.66, 7.70) (95% CI 7.62-7.70, 95% CI 7.66-7.74) | (7.42, 7.79) (95% CI 7.38-7.48, 95% CI 7.75-7.84) |
| LLaVA-NeXT (F+C) | 8.17 (95% CI 8.07-8.27) | (8.16, 7.75) (95% CI 8.09-8.22, 95% CI 7.67-7.84) | (7.59, 6.96) (95% CI 7.53-7.67, 95% CI 6.88-7.06) | 7.68 (95% CI 7.61-7.77) | (7.60, 7.58) (95% CI 7.58-7.66, 95% CI 7.55-7.63) | (7.29, 7.59) (95% CI 7.26-7.36, 95% CI 7.55-7.65) |
| LLaVA-NeXT (F-C) | 8.12 (95% CI 8.03-8.23) | (8.15, 7.79) (95% CI 8.08-8.21, 95% CI 7.71-7.88) | (7.63, 6.99) (95% CI 7.57-7.70, 95% CI 6.91-7.09) | 7.67 (95% CI 7.59-7.75) | (7.60, 7.58) (95% CI 7.56-7.64, 95% CI 7.55-7.63) | (7.28, 7.58) (95% CI 7.24-7.33, 95% CI 7.53-7.64) |
| MiniGPT-4 (B) | 7.04 (95% CI 6.93-7.17) | (5.22, 1.57) (95% CI 5.09-5.37, 95% CI 1.46-1.75) | (5.01, 1.52) (95% CI 4.89-5.16, 95% CI 1.40-1.67) | 7.07 (95% CI 6.97-7.16) | (5.06, 2.60) (95% CI 4.97-5.20, 95% CI 2.43-2.76) | (4.80, 2.57) (95% CI 4.70-4.93, 95% CI 2.44-2.78) |
| MiniGPT-4 (F+C) | 7.63 (95% CI 7.50-7.73) | (6.32, 0.34) (95% CI 6.23-6.39, 95% CI 0.25-0.38) | (5.98, 0.18) (95% CI 5.93-6.09, 95% CI 0.09-0.17) | 7.49 (95% CI 7.40-7.57) | (5.70, 0.94) (95% CI 5.62-5.76, 95% CI 0.81-1.05) | (5.58, 0.89) (95% CI 5.51-5.65, 95% CI 0.74-0.98) |
| MiniGPT-4 (F-C) | 7.59 (95% CI 7.48-7.70) | (6.24, 0.22) (95% CI 6.15-6.33, 95% CI 0.16-0.27) | (5.94, 0.18) (95% CI 5.88-6.04, 95% CI 0.11-0.20) | 7.41 (95% CI 7.31-7.49) | (5.52, 0.93) (95% CI 5.44-5.59, 95% CI 0.74-0.97) | (5.53, 0.88) (95% CI 5.46-5.61, 95% CI 0.73-0.97) |

*Table 19.* Wilcoxon $p$-values Comparing Text (Experiment A) vs. Chain (Experiment B) Scores at L2 and L3.

| VLM | GPT-4o | | Claude 3.5 | |
|---|---|---|---|---|
| | L2 | L3 | L2 | L3 |
| CogVLM2 | 0.0000 | 0.0000 | 0.0000 | 0.0000 |
| LLaVA-NeXT | 0.0000 | 0.0000 | 0.0000 | 0.0000 |
| MiniGPT-4 | 0.0000 | 0.0000 | 0.0000 | 0.0000 |
| mPLUG-Owl2 | 0.0000 | 0.0000 | 0.0000 | 0.0000 |
| Qwen-VL-Chat | 0.0000 | 0.0000 | 0.0000 | 0.0000 |
| InternVL2 | 0.0000 | 0.0000 | 0.0000 | 0.0000 |
| LLaVA-RLHF | 0.0000 | 0.0000 | 0.0000 | 0.0000 |
| LLaVA-CoT | 0.0000 | 0.0000 | 0.0000 | 0.0000 |

*Table 20.* Wilcoxon $p$-values: Baseline vs. Fine-tuned Models.

| VLM | GPT-4o | | | | | Claude 3.5 | | | | |
|---|---|---|---|---|---|---|---|---|---|---|
| | L1 | L2 | | L3 | | L1 | L2 | | L3 | |
| | *Text A* | *Text B* | *Chain B* | *Text B* | *Chain B* | *Text A* | *Text B* | *Chain B* | *Text B* | *Chain B* |
| LLaVA-NeXT | 0.0971 | 0.0005 | 0.0004 | 0.0188 | 0.0000 | 0.0001 | 0.0000 | 0.0000 | 0.0002 | 0.0000 |
| MiniGPT-4 | 0.0000 | 0.0000 | 0.0000 | 0.0000 | 0.0000 | 0.0000 | 0.0000 | 0.0000 | 0.0000 | 0.0000 |
| mPLUG-Owl2 | 0.0000 | 0.0000 | 0.3373 | 0.0000 | 0.2391 | 0.0000 | 0.0000 | 0.1531 | 0.0000 | 0.4887 |
| Qwen-VL-Chat | 0.0021 | 0.0000 | 0.0000 | 0.0000 | 0.0000 | 0.0001 | 0.0000 | 0.0000 | 0.0000 | 0.0000 |
| LLaVA-CoT | 0.2295 | 0.0000 | 0.0000 | 0.0000 | 0.0000 | 0.0000 | 0.0000 | 0.0000 | 0.0000 | 0.0000 |

*Table 21.* Wilcoxon $p$-values: Fine-tuned with vs. without Chain Supervision (Experiment B, L2/L3).

| VLM | GPT-4o | | Claude 3.5 | |
|---|---|---|---|---|
| | L2 | L3 | L2 | L3 |
| LLaVA-NeXT | 0.2281 | 0.6085 | 0.9153 | 0.7214 |
| MiniGPT-4 | 0.0887 | 0.7374 | 0.3492 | 0.5091 |

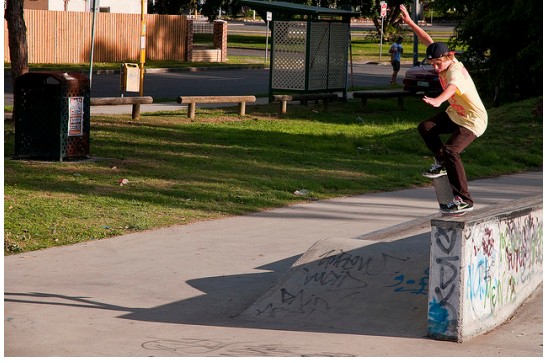

**Level 2 (Intervention) Question.** If it became nighttime, how might this affect the skateboarder's actions and safety?

**LLaVA-NeXT.** Night → Reduced Visibility → Increased Risk of Accidents. If it became nighttime, the skateboarder's actions and safety would be affected by the reduced visibility. The lack of natural light during nighttime makes it more difficult to see obstacles, judge distances, and maintain balance, increasing the risk of accidents. *(Chain: 9, Text: 9)*

**MiniGPT-4.** If it became nighttime, the skateboarder's actions and safety may be affected. The lack of natural light may make it harder to see obstacles and potential hazards, and the darkness may make it more difficult to navigate the park. Additionally, the reduced visibility may increase the risk of accidents and injuries. *(Chain: 0, Text: 8)*

*Figure 3.* Example 1, a skateboarder performing a trick. LLaVA-NeXT provides a correct causal chain and the textual answer supports the chain. MiniGPT-4 also provides a reasonable textual response but lacks a causal chain.

formats with predefined structures (Chen et al., 2024a; Li et al., 2025), CAGE specifically:

- Covers all three levels of Pearl's causal hierarchy systematically.
- Requires models to generate explicit causal structures instead of selecting from options.
- Uses real-world images with open-ended questions to challenge models to perform causal reasoning in naturalistic settings.
- Provides ground-truth causal chains aligned with formal causal theory.

These characteristics position CAGE as a complementary and more challenging benchmark for assessing generative causal reasoning capabilities in VLMs.

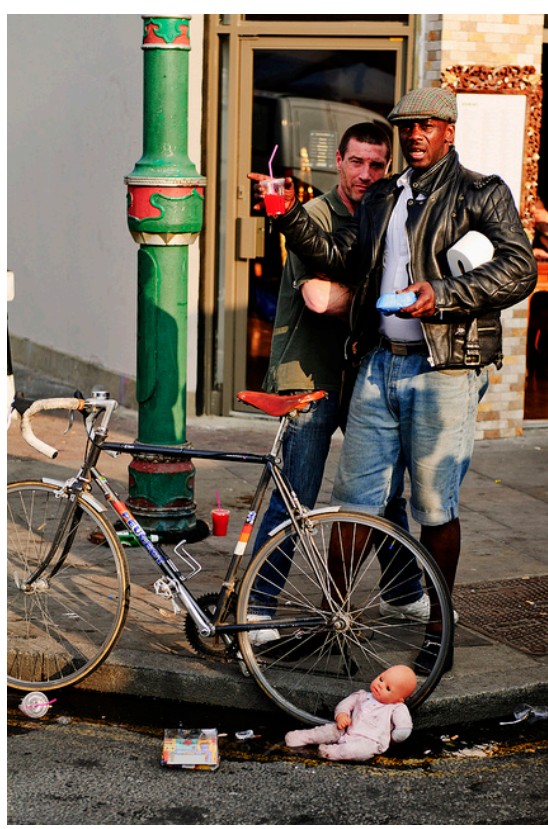

**Level 3 (Counterfactual) Question.** If a parade had passed by earlier, what might be visible on the street now?

**LLaVA-NeXT.** Parade → Street Decoration → Trash. If a parade had passed by earlier, the street would be decorated with parade-related items, such as banners, confetti, and possibly temporary barriers. After the parade, there might be trash left behind, which could be seen on the street now. *(Chain: 7, Text: 7)*

**MiniGPT-4.** Parade → Confetti/Streamers on ground. Man in suit → Confetti/Streamers on legs. The parade has caused confetti and streamers to be scattered on the ground, and some of them have stuck to the legs of a man in a suit who is walking towards the camera. *(Chain: 3, Text: 4)*

*Figure 4.* Example 2, two men standing on a sidewalk. LLaVA-NeXT provides a coherent causal chain but could have been more specific ("Confetti/Leftovers" instead of "Street Decoration"). Its textual answer aligns well with the chain and maintains a proper hypothetical framing. MiniGPT-4 generates a fragmented causal chain that fails to consistently use arrow notation and includes an unsupported link (Man in suit → Confetti/Streamers on legs). Its textual answer relies on hallucinated elements such as confetti and streamers, which are not visible in the image, and does not properly address the hypothetical scenario.

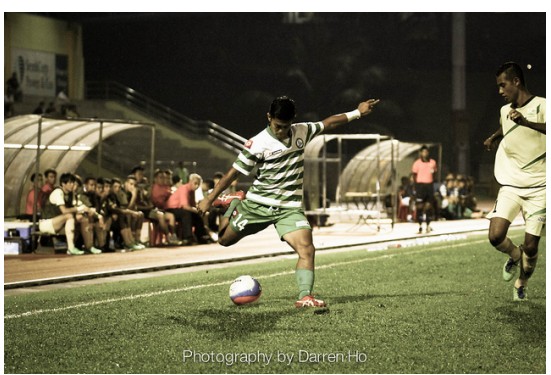

**Level 2 (Intervention) Question.** What would happen to the game if the floodlights were turned off suddenly?

**LLaVA-NeXT.** Floodlights → Illumination → Visibility → Continuation of Game. If the floodlights were turned off suddenly, the visibility on the field would be reduced, making it difficult for the players to see the ball and each other. This would likely cause the game to be interrupted or temporarily halted until the lights are turned back on. *(Chain: 6, Text: 8)*

**MiniGPT-4.** No floodlights → Darkness → Confusion. The sudden absence of floodlights causes the field to become dark, leading to confusion among the players. *(Chain: 7, Text: 7)*

*Figure 5.* Example 3, two men playing football. LLaVA-NeXT generates a causal chain with a flaw, as the final step reaches an incorrect conclusion. A more accurate chain would end with a game interruption or suspension. Its textual answer is stronger than the chain itself. MiniGPT-4 generates a decent causal chain with a supported textual answer. However, both lack detail about the broader consequences for the match.

## A.9. Evaluation on Object Hallucination Benchmarks

To assess whether our causal instruction fine-tuning impacts related visual grounding capabilities, we evaluate the fine-tuned models on two standard object hallucination benchmarks: MMHal-Bench (Sun et al., 2024) and POPE (Li et al., 2023b).

*Table 22.* Comparison of CAGE with other visual and multimodal reasoning benchmarks. 'Chain' indicates whether the dataset provides explicit causal chains. 'Causal Levels' specifies which of Pearl's Ladder levels are primarily targeted. Difficulty is an approximate indication based on reported model performance relative to humans or dataset design.

| Name | Size | Source | Reasoning Type | Causal Levels | Chain | Difficulty |
|---|---|---|---|---|---|---|
| SVRT (Fleuret et al., 2011) | 23 | Synthesized | Visual Pattern | Implicit/None | No | Easy |
| RAVEN (Zhang et al., 2019) | 70,000 | Synthesized | Relational/Analogical | Implicit/None | No | Easy |
| ARC (Chollet, 2019) | 1,000 | Synthesized | Abstract Visual | Implicit/None | No | Medium |
| DOPT (Webb et al., 2020) | 114,240 | Synthesized | Object Interaction/Physics | Primarily L2 | No | Easy |
| CLEVR (Johnson et al., 2017) | ~1,000,000 | Synthesized | Compositional QA | L1, some L2 | No | Easy |
| IQTest (Lu et al., 2024) | 228 | Mixed | Math/General Reasoning | Implicit/Mixed | No | Medium |
| MARVEL (Jiang et al., 2024b) | 770 | Web | Abstract Visual | Implicit/Mixed | No | Medium |
| MM-IQ (Cai et al., 2025) | 2,710 | Mixed | Abstraction/Reasoning | Implicit/Mixed | No | Medium |
| VQA v2.0 (Goyal et al., 2019) | 1,110,000 | Real | General QA | Primarily L1 | No | Easy/Medium |
| GQA (Hudson & Manning, 2019) | 22,000,000 | Real | Spatial/Logical QA | L1, some L2 | No | Medium |
| CELLO (Chen et al., 2024a) | 14,094 | Real | Causal Reasoning | L0, L1, L2, L3 | Yes (Predefined) | Hard |
| C-VQA (Zhang et al., 2023) | 6,144 | Real/Synthesized | Counterfactual Reasoning | L3 | No | Medium |
| MuCR (Li et al., 2025) | 12,000 | Synthetic | Cross-modal Causal | General Cause/Effect | No | Hard |
| CAGE (Ours) | 49,500 | Real | Causal Reasoning | L1, L2, L3 | Yes | Easy~Hard |

**MMHal-Bench Results** Table 23 presents the results on the MMHal-Bench benchmark. We see that our causal instruction fine-tuning does not significantly degrade general object hallucination performance. For most models, scores and hallucination rates remain comparable to their baselines. MiniGPT-4 shows a slight improvement in both metrics. Qwen-VL-Chat shows a slight decrease in hallucination rate while maintaining its score. This suggests that fine-tuning on the CAGE dataset primarily impacts causal reasoning capabilities and does not negatively interfere with existing visual grounding related to object hallucination, nor does it significantly improve it as a side effect.

*Table 23.* Baseline vs. Fine-tuned Performance on MMHal-Bench. Bold indicates fine-tuned models. Shaded rows group models.

| VLM | Average Score (Higher is Better) | Hallucination Rate (Lower is Better) |
|---|---|---|
| LLaVA-NeXT (Baseline) | 3.02 | 0.51 |
| **LLaVA-NeXT (Fine-tuned)** | **2.80** | **0.52** |
| mPLUG-Owl2 (Baseline) | 1.49 | 0.65 |
| **mPLUG-Owl2 (Fine-tuned)** | **1.43** | **0.67** |
| Qwen-VL-Chat (Baseline) | 2.66 | 0.48 |
| **Qwen-VL-Chat (Fine-tuned)** | **2.64** | **0.43** |
| LLaVA-CoT (Baseline) | 3.05 | 0.41 |
| **LLaVA-CoT (Fine-tuned)** | **2.80** | **0.43** |
| MiniGPT-4 (Baseline) | 1.68 | 0.65 |
| **MiniGPT-4 (Fine-tuned)** | **1.88** | **0.60** |

**POPE Results** The POPE results for the fine-tuned models across Random, Popular, and Adversarial prompt types are presented in Table 24. Similar to the MMHal-Bench results, the CAGE fine-tuning **does not significantly degrade** object hallucination performance on the POPE benchmark for the evaluated models. For most models, the scores (Accuracy, Precision, Recall, F1) remain largely comparable between their baseline and fine-tuned versions across all three POPE types. MiniGPT-4 shows an increase in Recall and F1 score after fine-tuning across all POPE types, while its Precision slightly decreases in Popular and Adversarial types. LLaVA-CoT shows increased Precision but decreased Recall and F1 score after fine-tuning. Qwen-VL-Chat and LLaVA-NeXT performance is largely stable. These results collectively suggest that the causal instruction fine-tuning, while targeting explicit causal chain generation, does not negatively interfere with general object grounding capabilities as measured by POPE, but also does not provide a consistent or substantial side benefit for improving general object hallucination. This reinforces that the causal modeling deficit probed by CAGE is distinct from general object hallucination and not necessarily addressed by training focused on textual fidelity.

*Table 24.* Baseline vs. Fine-tuned Performance on POPE (COCO 2014 val images).

| POPE Type | VLM | Accuracy | Precision | Recall | F1 Score |
|---|---|---|---|---|---|
| | LLaVA-NeXT (Baseline) | 0.92 | 0.95 | 0.88 | 0.91 |
| | **LLaVA-NeXT (Fine-tuned)** | **0.92** | **0.95** | **0.88** | **0.91** |
| | mPLUG-Owl2 (Baseline) | 0.86 | 0.89 | 0.82 | 0.85 |
| | **mPLUG-Owl2 (Fine-tuned)** | **0.87** | **0.88** | **0.85** | **0.87** |
| Random | Qwen-VL-Chat (Baseline) | 0.90 | 0.95 | 0.84 | 0.89 |
| | **Qwen-VL-Chat (Fine-tuned)** | **0.89** | **0.93** | **0.84** | **0.89** |
| | LLaVA-CoT (Baseline) | 0.88 | 0.95 | 0.80 | 0.87 |
| | **LLaVA-CoT (Fine-tuned)** | **0.87** | **0.98** | **0.76** | **0.85** |
| | MiniGPT-4 (Baseline) | 0.83 | 0.87 | 0.78 | 0.82 |
| | **MiniGPT-4 (Fine-tuned)** | **0.87** | **0.85** | **0.89** | **0.87** |
| | LLaVA-NeXT (Baseline) | 0.89 | 0.91 | 0.88 | 0.89 |
| | **LLaVA-NeXT (Fine-tuned)** | **0.89** | **0.90** | **0.88** | **0.89** |
| | mPLUG-Owl2 (Baseline) | 0.84 | 0.85 | 0.83 | 0.84 |
| | **mPLUG-Owl2 (Fine-tuned)** | **0.82** | **0.81** | **0.84** | **0.83** |
| Popular | Qwen-VL-Chat (Baseline) | 0.87 | 0.90 | 0.84 | 0.87 |
| | **Qwen-VL-Chat (Fine-tuned)** | **0.87** | **0.90** | **0.84** | **0.87** |
| | LLaVA-CoT (Baseline) | 0.86 | 0.92 | 0.80 | 0.86 |
| | **LLaVA-CoT (Fine-tuned)** | **0.86** | **0.95** | **0.76** | **0.84** |
| | MiniGPT-4 (Baseline) | 0.73 | 0.71 | 0.78 | 0.74 |
| | **MiniGPT-4 (Fine-tuned)** | **0.75** | **0.69** | **0.90** | **0.78** |
| | LLaVA-NeXT (Baseline) | 0.86 | 0.85 | 0.88 | 0.86 |
| | **LLaVA-NeXT (Fine-tuned)** | **0.86** | **0.84** | **0.88** | **0.86** |
| | mPLUG-Owl2 (Baseline) | 0.81 | 0.80 | 0.83 | 0.81 |
| | **mPLUG-Owl2 (Fine-tuned)** | **0.81** | **0.79** | **0.84** | **0.81** |
| Adversarial | Qwen-VL-Chat (Baseline) | 0.83 | 0.81 | 0.84 | 0.83 |
| | **Qwen-VL-Chat (Fine-tuned)** | **0.82** | **0.80** | **0.84** | **0.82** |
| | LLaVA-CoT (Baseline) | 0.85 | 0.88 | 0.80 | 0.84 |
| | **LLaVA-CoT (Fine-tuned)** | **0.85** | **0.93** | **0.76** | **0.83** |
| | MiniGPT-4 (Baseline) | 0.71 | 0.68 | 0.78 | 0.73 |
| | **MiniGPT-4 (Fine-tuned)** | **0.70** | **0.65** | **0.89** | **0.75** |

## A.10. Experimental Settings

This section provides detailed information regarding the experimental setup used for model fine-tuning and evaluation to ensure reproducibility of our results.

**Training Hardware**    Training of the fine-tuned models was conducted on a server equipped with $8 \times$ NVIDIA A40 GPUs (48GB each). This utilized distributed data parallelism for efficient training.

**Fine-tuning Configuration**    For causal instruction fine-tuning on the CAGE dataset, we adapted models using efficient methods appropriate for each VLM architecture to the new task while preserving pre-trained knowledge. Parameter-efficient methods, such as LoRA (Hu et al., 2022), were applied where suitable for the architecture. The AdamW (Loshchilov & Hutter, 2019) optimizer was used for all fine-tuning. Hyperparameter settings for each model were generally based on configurations recommended by their original projects or standard practices for fine-tuning models of their type. Note that gradient clipping was only applied during fine-tuning for MiniGPT-4. Specific hyperparameters and the adaptation method used for each VLM are detailed in Table 25.

**Inference Configuration**    Model inference for evaluation on the 500 test images (both Experiment A and Experiment B) was performed using the configurations detailed in Table 26. Unless otherwise noted, default decoding strategies provided by the respective model libraries were utilized.

*Table 25.* Fine-tuning Hyperparameters and Adaptation Methods for Evaluated VLMs.

| VLM | Base LLM / Vision Encoder | Adaptation Method | LoRA | Learning Rate | Batch Size | Epochs | Warmup Steps | Weight Decay |
|---|---|---|---|---|---|---|---|---|
| LLaVA-NeXT 13B | Vicuna-13B / CLIP ViT-L/14 | LoRA | Rank: 64, $\alpha$: 128 | $1 \times 10^{-4}$ | 16 | 1 | 30 | 0.01 |
| mPLUG-Owl2 8.2B | LLaMA / CLIP ViT-L | LoRA | Rank: 128, $\alpha$: 256 | $1 \times 10^{-4}$ | 16 | 1 | 10 | 0.0 |
| Qwen-VL-Chat 7B | Qwen-7B / CLIP ViT-L | LoRA | Rank: 64, $\alpha$: 16 | $1 \times 10^{-5}$ | 8 | 1 | 5 | 0.1 |
| MiniGPT-4 13B | Vicuna-13B / ViT-G | N/A | N/A | $5 \times 10^{-5}$ | 4 | 3 | 100 | 0.02 |
| LLaVA-CoT 11B | Llama 3.1 / Llama 3.2-Vision | LoRA | Rank: 16, $\alpha$: 16 | $2 \times 10^{-4}$ | 8 | 1 | 5 | 0.01 |

*Table 26.* Inference Configuration for Evaluated VLMs.

| VLM | Decoding Strategy | Max Generation Length | Temperature | Top-p | Repetition Penalty |
|---|---|---|---|---|---|
| CogVLM2 19B | Greedy | 2048 | 0.5 | 0.7 | N/A |
| LLaVA-NeXT 13B | Greedy | 2000 | 0.0 | N/A | N/A |
| mPLUG-Owl2 8.2B | Sampling | 1000 | 0.7 | N/A | N/A |
| Qwen-VL-Chat 7B | Sampling | 1024 | 0.85 | 0.8 | 1.1 |
| MiniGPT-4 13B | Sampling | 600 | 0.1 | 0.9 | 1.0 |
| InternVL2 76B | Greedy | 1024 | 0.7 | 0.95 | 1.1 |
| LLaVA-RLHF 13B | Sampling | 512 | 0.2 | 0.6 | N/A |
| LLaVA-CoT 11B | Greedy | 500 | 0.6 | 0.9 | N/A |

**Software and Libraries**   Experiments were conducted using standard deep learning frameworks and libraries, including PyTorch (Paszke et al., 2019), Hugging Face Transformers (Wolf et al., 2020), and PEFT. Specific versions used were PyTorch 2.3.0, Transformers 4.51.3, and CUDA 12.4.

### A.11. Ablation Study: Impact of Chain Supervision

To investigate the specific impact of explicit causal chain supervision during the causal instruction fine-tuning process, we conducted an ablation study. This study aims to determine whether training on the ground truth causal chain annotations is necessary or sufficient to improve models' ability to generate explicit causal structures, or if the limitation lies deeper.

**Experimental Setup**   We selected two models from our main evaluation: LLaVA-NeXT 13B, which showed relatively strong baseline chain generation capabilities, and MiniGPT-4 13B, which exhibited a significant baseline Abstraction Gap with very low chain scores.

For this ablation, we trained these two models under three conditions:

1. **Baseline:** The original pre-trained model without any CAGE fine-tuning.

2. **Fine-tuned with Chains (FT w/ Chains):** Causal instruction fine-tuning using the full 5000-image CAGE dataset (45,000 Q&A pairs), training with a joint loss function that optimizes for both text response quality and the correctness of the generated causal chains for Level 2 and 3 questions. This corresponds to the fine-tuning performed for the main results presented in Section 4.2.

3. **Fine-tuned without Chains (FT w/o Chains):** Fine-tuning using the same 5000-image CAGE dataset and questions, but with the ground truth causal chain annotations for Level 2 and 3 questions removed from the training data. The models were trained using a text-only loss for all levels. This effectively treated L2 and L3 as standard causal VQA tasks without the explicit structured output requirement during training.

All other fine-tuning hyperparameters and procedures were kept consistent between the "FT w/ Chains" and "FT w/o Chains" conditions.

After fine-tuning, both the "FT w/ Chains" and "FT w/o Chains" models, along with their Baselines, were evaluated on the 500 validation images across both the Text-Only Probe (Experiment A, text only) and the Chain-Text Probe (Experiment B, text and chain), and evaluated by both GPT-4o and Claude 3.5 Sonnet automated judges, following the protocols described in Section 3.

**Detailed Results**   The detailed results of the ablation study for Experiment A and Experiment B are presented in Tables 27 and 28. These show the evaluations by both GPT-4o and Claude 3.5 Sonnet.

*Table 27.* Ablation Study: Experiment A (Text-Only Probe) Scores (0-10), evaluated by GPT-4o and Claude 3.5 Sonnet.

| | GPT-4o Evaluation | | | Claude 3.5 Sonnet Evaluation | | |
|---|---|---|---|---|---|---|
| **VLM Condition** | **Avg L1** | **Avg L2** | **Avg L3** | **Avg L1** | **Avg L2** | **Avg L3** |
| LLaVA-NeXT 13B (Baseline) | 7.85 | 8.29 | 8.29 | 8.05 | 7.87 | 7.58 |
| LLaVA-NeXT 13B (FT w/ Chains) | 7.99 | 8.50 | 8.40 | 8.28 | 8.11 | 7.76 |
| LLaVA-NeXT 13B (FT w/o Chains) | 7.92 | 8.45 | 8.38 | 8.27 | 8.12 | 7.75 |
| MiniGPT-4 13B (Baseline) | 6.75 | 7.24 | 6.73 | 7.44 | 7.04 | 6.63 |
| MiniGPT-4 13B (FT w/ Chains) | 7.31 | 8.19 | 8.11 | 7.87 | 7.70 | 7.45 |
| MiniGPT-4 13B (FT w/o Chains) | 7.33 | 8.04 | 8.01 | 7.84 | 7.50 | 7.20 |

*Table 28.* Ablation Study: Experiment B (Chain-Text Probe) Scores (0-10), evaluated by GPT-4o and Claude 3.5 Sonnet.

| | GPT-4o Evaluation | | | Claude 3.5 Sonnet Evaluation | | |
|---|---|---|---|---|---|---|
| **VLM Condition** | **Avg L1 Text** | **Avg L2 (Text, Chain)** | **Avg L3 (Text, Chain)** | **Avg L1 Text** | **Avg L2 (Text, Chain)** | **Avg L3 (Text, Chain)** |
| LLaVA-NeXT 13B (Baseline) | 8.04 | (8.25, 7.95) | (7.85, 7.42) | 7.37 | (7.66, 7.70) | (7.42, 7.79) |
| LLaVA-NeXT 13B (FT w/ Chains) | 8.17 | (8.16, 7.75) | (7.59, 6.96) | 7.68 | (7.60, 7.58) | (7.29, 7.59) |
| LLaVA-NeXT 13B (FT w/o Chains) | 8.12 | (8.15, 7.79) | (7.63, 6.99) | 7.67 | (7.60, 7.58) | (7.28, 7.58) |
| MiniGPT-4 13B (Baseline) | 7.04 | (5.22, 1.57) | (5.01, 1.52) | 7.07 | (5.06, 2.60) | (4.80, 2.57) |
| MiniGPT-4 13B (FT w/ Chains) | 7.63 | (6.32, 0.34) | (5.98, 0.18) | 7.49 | (5.70, 0.94) | (5.58, 0.89) |
| MiniGPT-4 13B (FT w/o Chains) | 7.59 | (6.24, 0.22) | (5.94, 0.18) | 7.41 | (5.52, 0.93) | (5.53, 0.88) |

**Analysis**   The results from the ablation study provide critical insights into the role of explicit chain supervision during fine-tuning for improving VLM causal reasoning, particularly the ability to generate structured causal outputs. Tables 27 and 28 allow for a comparison across fine-tuning conditions and evaluation judges.

In Experiment A (Text-Only Probe, Table 27), both fine-tuning variants (with and without chains) generally lead to improved text scores for both LLaVA-NeXT and MiniGPT-4 across all levels compared to their baselines. This indicates that training on the CAGE dataset is effective in improving the models' ability to generate plausible causal language responses in a text-only format even without explicit chain supervision. The scores for "FT w/ Chains" and "FT w/o Chains" are very close. This suggests that the presence of chain annotations during training does not alter the models' performance on a purely text generation task, even when that text is causal.

The most insightful results are from Experiment B (Chain-Text Probe, Table 28), which evaluates both text and chain generation. For MiniGPT-4, which has a significant baseline Abstraction Gap (low chain scores, e.g., L3 Chain baseline of 1.52 by GPT-4o, 2.57 by Claude 3.5), both fine-tuning variants resulted in a substantial **decrease** in chain generation scores compared to its baseline, across both L2 and L3 and both judges. For example, MiniGPT-4's L3 Chain score dropped to around 0.18 (GPT-4o) / 0.88 (Claude 3.5) after fine-tuning, regardless of whether chain supervision was included. Furthermore, the chain scores for MiniGPT-4 fine-tuned *with* chain supervision are remarkably similar to those fine-tuned *without* chain supervision. This strongly suggests that for MiniGPT-4, simply providing explicit chain annotations during training does not help it to effectively learn to generate causal chains; in fact, the fine-tuning process seems to disrupt its baseline (albeit poor) chain generation capability in this structured output setting.

For LLaVA-NeXT, which has high baseline chain scores, fine-tuning (with or without chains) maintained relatively high chain scores, although GPT-4o showed a slight decrease after fine-tuning. There is a negligible difference in chain scores between the "FT w/ Chains" and "FT without Chains" conditions for LLaVA-NeXT across both judges and levels. This indicates that for a model already capable of generating chains, the explicit chain supervision in the training data does not provide a substantial additional benefit for this task within this fine-tuning setup.

The text scores in Experiment B generally improved with fine-tuning for MiniGPT-4, similar to Experiment A, but showed mixed results for LLaVA-NeXT. However, the primary insight from this ablation is the impact on chain generation.

This study provides empirical evidence supporting our conclusion that the Abstraction Gap is a fundamental limitation for many VLMs. The fact that MiniGPT-4's ability to generate chains did not improve even with explicit chain supervision, and that the performance of models fine-tuned with and without chain supervision is so similar for chain generation, suggests that the difficulty lies deeper than a simple lack of training data formatted with explicit chains. It implies challenges in the models' architecture or pre-training that limit their capacity to effectively learn and externalize structured causal representations through standard fine-tuning frameworks. We note that this ablation study was limited to two VLM architectures due to computational constraints. While a broader study across all fine-tuned models would provide more evidence, the consistent pattern observed in representative strong (LLaVA-NeXT) and weak (MiniGPT-4) models suggests that the limited impact of explicit chain supervision on chain generation is a general challenge within this fine-tuning framework. This reinforces that the Abstraction Gap is a potentially fundamental limitation.

