# OpenReview forum: "The Abstraction Gap in Vision-Language Causal Reasoning"
_ICML.cc/2026/Conference — ICML 2026 regular_

### Official Review · Reviewer_aE3s · 2026-03-05

**Soundness:** 2
**Presentation:** 1
**Significance:** 3
**Originality:** 3
**Overall Recommendation:** 4
**Confidence:** 4

**Summary:**

This paper studies the causal reasoning in the vision-language models (VLMs). The authors try to answer the question of whether current VLMs faithfully generate the causal rationale. To answer this question, the authors introduce the Text-Only Probe and the Chain-Text Probe to check the causal chain. They also introduce the Abstraction Gap (AG) metric to measure the plausibility-faithfulness distinction. Finally, they apply their metrics to the proposed CAGE benchmark and reveal the abstraction gap of current VLMs.

**Compliance With Llm Reviewing Policy:**

Affirmed.

**Final Justification:**

Thanks to the authors'  detailed rebuttal. I would like to suggest that they carefully revise their presentation (figures, tables) and fully explain why their newly added experiments enhance their benchmark's motivation. I won't stop if other reviewers champion this submission. I have increased my score to a weak accept (4).

**Key Questions For Authors:**

What is the motivation for making a new benchmark? Can’t the proposed method be applied to other benchmarks?

**Limitations:**

Please refer to the weaknesses part.

**Strengths And Weaknesses:**

Strengths:
1. The studied problem, the faithfulness of causal reasoning in VLMs, is practical and worth exploring.

2. The proposed benchmark is large and comprehensive, which I think is a core contribution.

3. The experiments are comprehensive. The authors test multiple VLMs.

Weaknesses:
1. I think the first apparent weakness is the logic of the abstract. Specifically, the authors directly introduce the faithfulness problem without introducing the gap of existing evaluation framework. Why can't the existing evaluation frameworks answer this question? I suggest that the authors add a sentence to demonstrate it.

2. The motivation of the benchmark is unclear. The core assumption or idea is: “if a model truly understands causality in visual scenes, it should be able to abstract its reasoning into a structural form”. However, I believe this assumption may not be reasonable. Given the web-scale of current training datasets, even if a VLM can perform causal abstraction, its generation may come from the correlations rather than true causal understanding.

3. The introduction section may be too long (3 pages) and has some structural problems. For example, the limitations of existing works are discussed in the fourth paragraph. I think it may be better to move it to the third or second paragraph.

4. The proposed benchmark CAGE is good, but some details can be added to the introduction. For example, the difference with existing benchmarks.

5. Figure 1 is not well drawn. I think it’s a combination of table and figure rather than a unified figure. The authors should try to unify the table and figure to better visualize their idea.

6. The content of the tables is too dense. For example, the table 5 and 6. I suggest that authors make the tables larger.

---

> ### Author Rebuttal · Authors · 2026-03-30
>
> We appreciate the reviewer's recognition of our benchmark's scope and the importance of the faithfulness problem. The concerns raised deserve careful engagement.
>
> The reviewer questions our core assumption: even if a model produces causal chains, might this reflect learned correlations rather than genuine understanding? This is a fair challenge, and we acknowledge that no benchmark can definitively prove "true understanding." What we can do is measure a gap that is robust across conditions and tracks real capability differences.
>
> Several findings support the diagnostic value of our approach. Level 3 questions require counterfactual reasoning about scenarios not depicted in the image; models must infer what would change, not recognize what is present. Fine-tuning on 45,000 chain-formatted examples does not close the gap for most models; if chain generation were pattern-matching, this exposure should transfer. It does not.
>
> To investigate whether the gap reflects difficulty with arrow notation, we conducted an experiment using comma-separated sequences:
>
> | Model | Arrow AG | Comma AG |
> |:------|:---------|:---------|
> | LLaVA-NeXT | 0.035 | 0.008 |
> | MiniGPT-4 | 0.700 | 0.991 |
> | mPLUG-Owl2 | 0.917 | 0.958 |
> | Qwen-VL-Chat | 0.501 | 0.400 |
>
> If chains were generated from surface correlations, a simpler format should not degrade performance. For MiniGPT-4 and mPLUG-Owl2, the gap widens. Qwen-VL-Chat improves modestly but retains a substantial gap. LLaVA-NeXT remains unaffected.
>
> We also evaluated Gemini 2.5 Flash, which achieves AG of -0.006, essentially eliminating the gap. This is important: if the gap were an artifact of benchmark design, all models should show similar patterns. They do not. Some fail (AG > 0.7), others succeed (AG ≈ 0). CAGE tracks real capability differences and can measure progress.
>
> Whether this constitutes "true understanding" remains a hard philosophical question. Our contribution is a diagnostic tool that measures something robust, format-independent, and variable across models.
>
> On benchmark motivation: existing benchmarks use selection formats where models choose among candidates. Selection tests recognition; CAGE tests generation. A model could identify correct chains without being able to produce them. The dual-probe methodology could adapt to other datasets, but they lack the image-grounded causal chains needed for our Chain-Text Probe. CAGE provides both methodology and data.
>
> On the abstract (W1), we will revise it to first state the evaluation gap: "Vision-language models produce fluent causal explanations, but current evaluations cannot distinguish linguistic plausibility from faithful causal reasoning."
>
> For the camera-ready, we will restructure the introduction (moving limitations earlier, reducing length), add explicit benchmark comparisons, redesign Figure 1 as a unified visualization, and reduce table density by moving details to the appendix.

---

> > ### Author Rebuttal · Reviewer_aE3s · 2026-04-01
> >
> > Thanks to the authors' response. First, I think the authors should present their rebuttal in a point‑by‑point format, with the key points highlighted in bold. The current version is hard for me to distinguish the main points immediately.
> >
> > In addition, I believe the structural and presentation problems, such as the abstract, introduction, and figure, should not be present in the initial submission. Therefore, I will maintain my current score.

---

> > > ### Author Response · Authors · 2026-04-01
> > >
> > > We have restructured our response for clarity.
> > >
> > > **W2/Key Question: Core assumption and benchmark motivation.**
> > >
> > > The reviewer's central concern: could causal chains reflect learned correlations rather than genuine understanding? This is a fair challenge. We acknowledge that no benchmark can definitively prove "true understanding." What we can do is measure a gap that is robust across conditions and tracks real capability differences. We present converging evidence, including **two new experiments**:
> > >
> > > **(1) Fine-tuning (in paper):** Training on 45,000 chain-formatted examples does not close the gap. If chain generation were pattern-matching, this exposure should transfer. It does not.
> > >
> > > **(2) Format ablation (new):** We tested comma-separated sequences instead of arrows:
> > >
> > > | Model | Arrow AG | Comma AG |
> > > |:------|:---------|:---------|
> > > | LLaVA-NeXT | 0.035 | 0.008 |
> > > | MiniGPT-4 | 0.700 | 0.991 |
> > > | mPLUG-Owl2 | 0.917 | 0.958 |
> > > | Qwen-VL-Chat | 0.501 | 0.400 |
> > >
> > > **MiniGPT-4 and mPLUG-Owl2 worsen; Qwen-VL-Chat improves modestly but retains a gap; LLaVA-NeXT is unaffected.** The limitation is not format-specific.
> > >
> > > **(3) Gemini 2.5 Flash (new):** Achieves AG = -0.006, eliminating the gap. **If the gap were a benchmark artifact, all models should show similar patterns. They do not.** Some fail (AG > 0.7), others succeed (AG ≈ 0).
> > >
> > > Whether this constitutes "true understanding" remains a hard philosophical question. Our contribution is a diagnostic tool that measures something robust, format-independent, and variable across models.
> > >
> > > On benchmark motivation: existing benchmarks use selection (recognition); CAGE requires generation (production). A model could identify correct chains without producing them. The dual-probe methodology could adapt to other datasets, but they lack the image-grounded causal chains for our Chain-Text Probe. CAGE provides both methodology and data.
> > >
> > > **W1: Abstract.** We will revise to state the evaluation gap first: "Vision-language models produce fluent causal explanations, but current evaluations cannot distinguish linguistic plausibility from faithful causal reasoning."
> > >
> > > **W3-W6: Presentation.** We will restructure the introduction (moving limitations earlier), add benchmark comparisons, redesign Figure 1 as a unified visualization, and reduce table density.
> > >
> > > Our **new experiments** directly address W2 with empirical evidence. We hope the reviewer will engage with whether these findings resolve the substantive concern.

---

### Official Review · Reviewer_svnX · 2026-03-08

**Soundness:** 3
**Presentation:** 3
**Significance:** 3
**Originality:** 3
**Overall Recommendation:** 4
**Confidence:** 3

**Summary:**

A dual-probe method is proposed to evaluate VLMs textual explanations (representing plausibility) versus the explicit generation of symbolic causal chains (representing faithfulness). Causal Abstraction Gap Evaluation (CAGE), a benchmark of 49500 questions  across 5500 images is introduced to test whether VLMs possess genuine causal understanding based on the dual-probe method. The paper concludes that most VLMs suffer from a severe abstraction gap, generating fluent text but failing to produce valid causal chains even after fine-tuning.

**Compliance With Llm Reviewing Policy:**

Affirmed.

**Final Justification:**

The added experiments in rebuttal addresses most of my concerns. Despite some concerns remain, I believe that the paper after revision offers insights that others could build on. I would like to increase my score to "weak accept".

**Key Questions For Authors:**

- The human validation is provided for only two models, is it possible to provide the complete human validation?
- Is there an ablation for text-then-chain versus chain-then-text?

**Limitations:**

- Provide a diagnosis of the high blank output rate.
- Verify the hypothesis why LLaVA-NeXT stands out.

**Strengths And Weaknesses:**

Strengths:
- Novel evaluation paradigm. The explicit causal chain generation measures structural understanding more so than just pattern matching.
- The creation of a large-scale, open-ended benchmark grounded in Pearl's causal hierarchy is a great contribution for the community.
- The empirical observation that perceptual grounding (object hallucination) is independent of structural grounding (causal abstraction) is a valuable and insightful contribution.
- Thorough experiment results provided.

Weaknesses:
- The paper's core method rests on the assumption that because humans construct a preverbal representation of relationships before formulating language, VLMs should also behave similarly, and forced to output structural chains *before* text. This is not necessarily true, and potentially contradicts to auto-regressive generation. LLMs/VLMs use generated language as a computational scratchpad, if forced to do a generation of a symbolic chain before any natural language text, it might not have enough tokens to reason. A model generating a valid chain *after* textual reasoning does not automatically invalidate its causal capabilities.
- The "blank outputs" are attributed to models failure in structural abstraction. Some models for example are reported to produce thousands of blank responses. While outputting incorrect causal chain is within expectation, I find this many blank outputs concerning, are they caused by immediate EoS tokens? Can this be due to an alignment issue, OOD issue, or engineering/harness issue?
- The method rigidly requires causal chains to be formatted using specific arrow notation (e.g., A -> B -> C). Is it possible for a VLM to possess causal understanding but fail the strict syntactic constraints? Some ablations to verify that format is not the cause of causal chain generation failure are needed.
- The paper observes that LLaVA-NeXT is the only model to achieve a near-zero AG and hypothesizes this is due to instruction tuning on diagram datasets like AI2D and ChartQA. However, the study lacks the controlled ablation experiments necessary to confirm this specific hypothesis.

---

> ### Author Rebuttal · Authors · 2026-03-30
>
> We appreciate the reviewer's careful engagement with our methodology, particularly the questions about prompt ordering and blank outputs. The concerns are substantive, and we have conducted new experiments to address them.
>
> The reviewer's concern about chain-before-text ordering (W1) is well-taken. We ran a text-then-chain ablation:
>
> | Model | Chain-Text AG | Text-Chain AG |
> |:------|:--------------|:--------------|
> | LLaVA-NeXT | 0.035 | 0.013 |
> | MiniGPT-4 | 0.700 | 0.537 |
> | mPLUG-Owl2 | 0.917 | 0.742 |
> | Qwen-VL-Chat | 0.501 | 0.144 |
>
> All four models improve with text-first ordering, confirming that prompt design affects performance. Qwen-VL-Chat shows the largest change (0.501 → 0.144), suggesting partial structural capability that is more accessible with self-scaffolding. However, MiniGPT-4 and mPLUG-Owl2 retain substantial gaps (0.537 and 0.742) even with text-first ordering, indicating these models lack structural capability regardless of scaffolding.
>
> We designed CAGE to measure whether models can produce structure without first verbalizing, because this tests whether structural representations exist independently of language generation. Text-first ordering tests a different question: whether models can formalize what they have already articulated. Both are valid diagnostics. We will report text-first results in the camera-ready and discuss scaffolding-dependent versus scaffolding-independent abstraction.
>
> Our framing does not assume VLMs should work like human cognition. We test capability, not mechanism: can models produce structure when asked?
>
> On blank outputs (W2), we analyzed all outputs using a failure mode taxonomy:
>
> | Model | Format-Compliant | Non-Response | Text-Only | Other |
> |:------|:-----------------|:-------------|:----------|:------|
> | LLaVA-NeXT | 89.0% | 0.0% | 1.1% | 9.9% |
> | LLaVA-RLHF | 9.0% | 76.4% | 11.4% | 3.3% |
> | CogVLM2 | 1.6% | 16.7% | 81.3% | 0.4% |
> | MiniGPT-4 | 10.9% | 0.0% | 61.1% | 28.0% |
>
> Blank rates vary dramatically under identical conditions: LLaVA-RLHF produces 76.4% blanks while LLaVA-NeXT produces none. If blanks reflected harness issues, we would expect similar rates. The concentration in LLaVA-RLHF suggests alignment training may induce output suppression when encountering unfamiliar structural formats.
>
> On format rigidity (W3), we evaluated four models using comma-separated sequences instead of arrows:
>
> | Model | Arrow AG | Comma AG |
> |:------|:---------|:---------|
> | LLaVA-NeXT | 0.035 | 0.008 |
> | MiniGPT-4 | 0.700 | 0.991 |
> | mPLUG-Owl2 | 0.917 | 0.958 |
> | Qwen-VL-Chat | 0.501 | 0.400 |
>
> If arrow syntax were the barrier, commas should help. For MiniGPT-4 and mPLUG-Owl2, performance worsens. Qwen-VL-Chat improves modestly but retains a gap. Models that fail with arrows do not succeed with commas.
>
> On the LLaVA-NeXT hypothesis (W4), we evaluated LLaVA-1.5, which shares the same architecture but lacks diagram training. LLaVA-1.5 achieves AG of 0.051, nearly identical to LLaVA-NeXT (0.035). The base architecture itself supports structural generation. The more interesting question is why LLaVA-RLHF fails (AG = 0.843) despite sharing this architecture. We hypothesize that RLHF optimization for fluent text may suppress structured output, though this requires further investigation.
>
> On human validation (Q1), we will extend to a stratified sample across all eight models in the camera-ready.

---

> > ### Author Rebuttal · Reviewer_svnX · 2026-04-03
> >
> > W1: While I agree that  text-chain AG tests whether models can formalize what they have already articulated, I think it's arguable whether it also tests causal abstraction ability. And I'm not fully convinced that chain-text AG tests whether structural representations exist *independently* of language generation, because the structural representations still need to be presented via some form of language generation. That being said, I think the added experiment is very informative, and does address most of my concern.
> >
> > W2: The added experiment addresses my concern.
> >
> > W3: This addresses most of my concern. Does using comma AG make fine-tuning more effective?
> >
> > W4: Though the question remains, I'm glad to see that more insights are provided.
> >
> > Q1: Concern addressed.

---

> > > ### Author Response · Authors · 2026-04-03
> > >
> > > We are pleased that most concerns are now resolved. We respond to the remaining questions below.
> > >
> > > **W3 follow-up: Does comma format make fine-tuning more effective?**
> > >
> > > Our evidence suggests it would not. The key observation: comma format does not improve zero-shot performance; MiniGPT-4 and mPLUG-Owl2 actually worsen. If the barrier were format familiarity, simpler notation should help; it does not. Combined with the fine-tuning failure (45,000 arrow examples), this suggests the barrier lies in structural abstraction itself, not in notation. We will note comma fine-tuning as future work.
> > >
> > > **W1 clarification:**
> > >
> > > The reviewer raises a valid point: both chain-first and text-first involve language generation, so claiming chain-first tests "independence from language generation" is imprecise. We should clarify.
> > >
> > > What chain-first actually tests is whether models can *directly serialize* a structured causal representation without first articulating reasoning in prose. What text-first tests is whether models require prose generation as an *intermediate step* before producing structure. Both involve language output, but they probe different representational access: direct externalization versus mediated via verbalization.
> > >
> > > The empirical finding is informative regardless: models succeeding only with text-first appear to need verbalization as scaffolding. We will sharpen this framing in the camera-ready.
> > >
> > > To address the reviewer's concerns, we conducted four new experiments during rebuttal: text-chain ablation, comma format ablation, LLaVA-1.5 evaluation, and Gemini 2.5 Flash evaluation. The Gemini result (AG = -0.006) validates that CAGE measures real capability differences; the gap is closable, not an artifact.
> > >
> > > We thank the reviewer for the constructive dialogue and believe these experiments address the core methodological concerns raised.

---

### Official Review · Reviewer_iaKK · 2026-03-11

**Soundness:** 3
**Presentation:** 3
**Significance:** 3
**Originality:** 3
**Overall Recommendation:** 4
**Confidence:** 3

**Summary:**

This paper investigates whether plausible answers generated by vision-language models truly reflect faithful structural causal understanding. The authors propose a dual-probe evaluation framework: a Text-Only Probe that measures plausible verbal reasoning, and a Chain-Text Probe that requires the model to first generate an explicit causal chain and then explain it in text. They introduce the Abstraction Gap (AG) metric to quantify the relative drop from text performance to chain-generation performance, and construct the CAGE benchmark with 49,500 questions over 5,500 COCO images spanning Pearl’s three causal levels. The main finding is that most evaluated VLMs achieve reasonably strong text scores yet fail badly at generating minimal causal chains, yielding large AG values; moreover, fine-tuning on 45,000 chain-annotated examples generally does not close this gap.

**Compliance With Llm Reviewing Policy:**

Affirmed.

**Final Justification:**

I appreciate the authors’ rebuttal and the additional experiments. The paper studies an important question: whether VLMs truly have structural causal understanding, rather than only generating plausible causal language. I find the proposed evaluation framework meaningful, and the benchmark and experiments are generally strong.

The rebuttal addresses my main concerns reasonably well. In particular, the additional results on alternative output formats suggest that the observed gap is not merely caused by the arrow-chain representation, and the added evaluation on newer models further strengthens the paper. I still think the paper should be careful not to overclaim about the absence of internal causal reasoning, since the current evidence more directly supports a weakness in explicit structured causal expression. However, this does not change my overall positive assessment.

**Key Questions For Authors:**

1.	The paper mentions a three-stage data validation process. Could the authors provide more detail on how each stage was conducted and how these steps ensured the quality and reliability of the synthetic dataset?

2.	Have the authors evaluated the Abstraction Gap (AG) of large proprietary VLMs? Such results would help clarify whether the observed gap is specific to the open-source models considered here or reflects a broader limitation of current VLMs.

3.	Could the authors provide representative failure cases or examples spanning different values of C(M,d)? In particular, is performance primarily limited by difficulty generating the required causal-chain format, or by a deeper failure of causal reasoning?

4.	LLaVA-CoT shows unusually low text scores on the Text-Only Probe (e.g., 2.93 at Level 2 under GPT-4o evaluation in Table 1), yet improves substantially to 7.26 (Table 2) on the Chain-Text Probe. How do the authors explain this reversal? Does the chain-generation prompt scaffold better reasoning, or does this suggest substantial prompt sensitivity?

**Limitations:**

yes

**Strengths And Weaknesses:**

**Strengths**

1. The paper asks a meaningful question. The distinction between plausible explanation and faithful structural reasoning is important, especially for multimodal systems that may sound convincing without actually representing causal structure. The proposed AG metric is simple, interpretable, and well aligned with the paper’s central claim.

2. The benchmark design is also a great contribution. CAGE goes beyond multiple-choice or selection-style causal benchmarks by requiring explicit generation of lightweight causal chains, which is a cleaner test of whether a model can externalize structure rather than merely recognize it. This makes the setup more diagnostic than standard evaluation formats.

3. The empirical results are interesting. The paper reports that seven of eight tested models have AG above 0.50, with some models showing near-complete collapse in chain generation despite strong textual performance. The fine-tuning results are also useful: they suggest that simple supervised exposure to chain-formatted outputs is often insufficient to induce the missing capability.

**Weaknesses**

1.	Limited model coverage. The evaluation is restricted to open-source VLMs, so it remains unclear whether the same conclusions hold for large proprietary models. Including stronger closed-source systems would provide a more complete picture of the abstraction gap and help clarify whether this limitation is specific to current open models or more fundamental.

2.	Potential confounding from output format. It is not yet fully clear whether failure on the Chain-Text Probe cleanly reflects a lack of causal abstraction, rather than difficulty with a particular output format or prompting interface. Although the paper argues that lightweight arrow chains constitute a minimal structural representation, some models may be disproportionately sensitive to formatting conventions, ordering constraints, or instruction-following details.

3.	Lack of qualitative examples across score ranges. The paper does not provide representative examples of causal-reasoning chains receiving different score ranges (e.g., 8–10, 6–7, 3–5, 1–2). As a result, it is difficult for readers to develop an intuitive sense of what these scores correspond to in terms of causal-reasoning quality.

---

> ### Author Rebuttal · Authors · 2026-03-30
>
> We appreciate the reviewer's positive assessment and address each concern below.
>
> On proprietary and recent models (W1/Q2), we evaluated Gemini 2.5 Flash and Qwen3-VL:
>
> | Model | Text | Structure | AG |
> |:------|:-----|:----------|:-----|
> | Gemini 2.5 Flash | 8.60 | 8.46 | -0.006 |
> | Qwen3-VL | 7.80 | 5.74 | 0.313 |
>
> The contrast is informative. Qwen3-VL, despite being a recent and capable open-source model, retains a moderate gap. Gemini 2.5 Flash essentially eliminates it (AG = -0.006). This suggests the limitation is not inherent to VLM architectures but can be overcome with sufficient capability or training. CAGE can thus track progress as models improve, validating its use as a diagnostic tool for measuring capability rather than merely exposing failures.
>
> On format confounding (W2), we tested four models using comma-separated sequences instead of arrows:
>
> | Model | Arrow AG | Comma AG |
> |:------|:---------|:---------|
> | LLaVA-NeXT | 0.035 | 0.008 |
> | MiniGPT-4 | 0.700 | 0.991 |
> | mPLUG-Owl2 | 0.917 | 0.958 |
> | Qwen-VL-Chat | 0.501 | 0.400 |
>
> If the gap reflected format-specific difficulty, commas should help. For MiniGPT-4 and mPLUG-Owl2, AG actually increases. Qwen-VL-Chat improves modestly but retains a substantial gap. Models that fail with arrows do not succeed with commas.
>
> On qualitative examples (W3/Q3), here is a concrete case. For a Level 2 intervention question about a living room scene, LLaVA-NeXT produced:
>
> > "Closed Curtains → Reduced Natural Light → Dimmer Room. If the curtains were drawn closed, the room's lighting would become dimmer due to the reduced natural light entering the room." (score 8)
>
> MiniGPT-4 produced:
>
> > "The curtains being drawn closed would not significantly affect the room's lighting. The room's natural light would be blocked, but any artificial light sources in the room, such as lamps or overhead lighting, would still be visible." (score 0, no chain)
>
> The contrast is clear: MiniGPT-4 produces fluent text but omits the requested structure entirely. We will add a comprehensive appendix with examples spanning score ranges.
>
> On data validation (Q1), our three stages were: (1) Level verification using an LLM to classify each question's target causal level against Pearl's hierarchy; (2) Visual answerability review of 600 randomly sampled questions to confirm they were answerable from the image; (3) Chain consistency review of 600 causal chains to verify logical coherence and sufficient causal links. We will expand on this in the camera-ready.
>
> On the LLaVA-CoT anomaly (Q4), the reviewer notes this model scores 2.93 on Text-Only but 7.26 on Chain-Text (text component). We believe requiring chain generation first scaffolds the model's reasoning. LLaVA-CoT was trained with chain-of-thought supervision, so under Text-Only, it produces verbose but poorly organized output. Under Chain-Text, the structure requirement organizes the reasoning, producing better subsequent text. Crucially, this does not mean LLaVA-CoT succeeds at structural abstraction; its chain scores remain low (below 3.0). The prompting format affects how the model organizes output, illustrating that prompt design interacts with training in model-specific ways.

---

> > ### Author Rebuttal · Reviewer_iaKK · 2026-04-05
> >
> > Thanks to the authors for addressing my concerns. I will maintain my score in support of the paper.

---

> > > ### Author Response · Authors · 2026-04-06
> > >
> > > We thank the reviewer for their support and thoughtful questions. We will expand the appendix with additional examples spanning score ranges as discussed.

---

### Official Review · Reviewer_WPk6 · 2026-03-13

**Soundness:** 3
**Presentation:** 4
**Significance:** 3
**Originality:** 3
**Overall Recommendation:** 5
**Confidence:** 5

**Summary:**

This paper investigates whether vision-language models (VLMs) that produce fluent causal explanations actually possess structural causal understanding. The authors propose that current evaluations mostly measure plausibility (linguistic quality) rather than faithfulness (actual reasoning structure). To study this, they introduce Dual-Probe evaluation where the text-only probe to answer directly and chain-text probe requires models to first generate an explicit causal chain before producing a textual answer. They also introduced a metric called Abstraction Gap (AG) which measures the gap between plausible explanations and causal structure. They constructed a new benchmark called CAGE to evaluate this gap based on Pearl’s ladder of causation, including 5,500 COCO images and 49,500 question-answer pairs. The authors evaluate 8 open-source VLMs in various sizes, and find 7 of 8 models have major mismatch between text generation and chain generation (AG > 0.5). Finetuning does not improve this at all. They conclude that most of the SOTA VLMs produce plausible causal language but lack structural causal abstraction capability.

**Compliance With Llm Reviewing Policy:**

Affirmed.

**Final Justification:**

The paper tackles an important question about causal reasoning in VLMs with a novel and insightful evaluation framework, and the rebuttal has adequately addressed my concerns, so I support acceptance.

**Key Questions For Authors:**

1. Given these differences in evaluation criteria, how do the authors justify that the two scoring scales are comparable enough to compute AG directly?
2. How do the experiments distinguish between absence of internal causal structure and difficulty externalizing that structure into symbolic chain notation?
3. Have the authors tested alternative structural representations (e.g., step-by-step natural language reasoning, bullet lists, or structured JSON outputs) to verify that the observed gap is not specific to the arrow-chain output format?

**Limitations:**

Yes

**Strengths And Weaknesses:**

Strengths
1. The paper targets a fundamental issue in multimodal reasoning whether models actually reason causally or merely produce convincing language. This is a very important field to understand whether AI reasoning is truly human-level yet.
2. I really appreciate the authors' connection to cognitive science in evaluating AI models, especially their motivation from Pearl’s causal hierarchy.
3. The AG metric is simple and intuitive, and it clearly captures the phenomenon that models perform well in text but fail in structured reasoning. The authors also provide human validation of this metric, further making it more convincing.
4. Through extensive experiments, the authors provide sufficient evidence to show that large models still fail at simple causal chains and fine-tuning does not fix the problem.

Weaknesses
1. The AG metric compares text and chain scores as if they were commensurate, but the underlying scoring rubrics differ substantially. Text responses are evaluated for plausibility and completeness, whereas chains are evaluated for structural validity and minimum-link requirements. They can potentially in a different scale given that their rubrics are different. It is like comparing the scores of a hard test and an easy one. Without stronger calibration or normalization, it is unclear whether the reported Abstraction Gap reflects a true reasoning gap or simply the increased difficulty of satisfying a stricter output format.
2. The authors partially address the concern that chain failure is merely a formatting issue by fine-tuning on chain-annotated data and by ablating chain supervision. Still, these experiments do not fully rule out the possibility that some structural knowledge is represented implicitly but cannot be reliably externalized in the specific linear arrow-chain format used here. The current evidence supports a claim about failure of explicit structural expression, but a stronger claim about absence of internal structural reasoning seems overstated.
3. I think the authors should test more recent models as well like InternVL3, Qwen2.5-VL, GPT-5 and so on. But I think this point is rather minor compared to the previous two.

---

> ### Author Rebuttal · Authors · 2026-03-30
>
> We appreciate the reviewer's careful attention to methodological details. We address each concern with new evidence.
>
> On score calibration (W1/Q1), the reviewer asks whether AG might reflect rubric difficulty rather than capability. We think not. Under the same chain rubric, scores range from 7.95 (LLaVA-NeXT) to 0.09 (CogVLM2). If the rubric were simply harder, all models should score low. Instead, some achieve high chain scores while others approach zero under identical conditions. This variation reflects capability, not task difficulty. For the camera-ready, we will add explicit calibration analysis: score distributions per probe, inter-judge agreement per rubric, and variance analysis conditioned on model capability.
>
> On format-specific externalization (W2/Q2/Q3), we tested four models using comma-separated sequences (e.g., A, B, C) instead of arrows:
>
> | Model | Arrow AG | Comma AG |
> |:------|:---------|:---------|
> | LLaVA-NeXT | 0.035 | 0.008 |
> | MiniGPT-4 | 0.700 | 0.991 |
> | mPLUG-Owl2 | 0.917 | 0.958 |
> | Qwen-VL-Chat | 0.501 | 0.400 |
>
> If arrow syntax were the barrier, commas should help. For MiniGPT-4 and mPLUG-Owl2, performance actually worsens. Qwen-VL-Chat improves modestly but retains a substantial gap. LLaVA-NeXT remains unaffected. Models that fail with arrows do not succeed with commas.
>
> The reviewer also asked about step-by-step natural language. We note that the Text-Only Probe already elicits natural language causal explanations, and models succeed there (scores 5.0-7.8) while failing at explicit structure. This suggests the difficulty lies not in the notation but in the abstraction into relational form. Models describe causal relationships fluently; they struggle to organize those relationships into any structured representation.
>
> On recent models (W3), we evaluated Qwen3-VL and Gemini 2.5 Flash:
>
> | Model | Text | Structure | AG |
> |:------|:-----|:----------|:-----|
> | Qwen3-VL | 7.80 | 5.74 | 0.313 |
> | Gemini 2.5 Flash | 8.60 | 8.46 | -0.006 |
>
> The contrast is striking. Qwen3-VL maintains a moderate gap (0.313), indicating that the limitation persists in recent open-source models. Gemini 2.5 Flash, however, essentially eliminates the gap (AG = -0.006). This demonstrates that the Abstraction Gap is not an inevitable property of VLM architectures but a limitation that can be overcome. CAGE thus serves not only as a diagnostic of current weaknesses but as a benchmark for tracking progress. We find it encouraging that at least one model has closed the gap, and we will discuss the implications in the camera-ready.

---

> > ### Author Rebuttal · Reviewer_WPk6 · 2026-04-01
> >
> > I appreciate the rebuttal. It has fully resolved my concerns.

---

> > > ### Author Response · Authors · 2026-04-06
> > >
> > > We thank the reviewer for their thorough evaluation and constructive engagement throughout this process. We will incorporate the calibration analysis and recent model results into the camera-ready version.

---

### Decision · Program_Chairs · 2026-04-30

**Decision:**

Accept (regular)

**Comment:**

The paper proposes a dual-probe evaluation framework consisting of a Text-Only Probe that measures plausible verbal reasoning, and a Chain-Text Probe that requires the model to first generate an explicit causal chain and then explain it in text. The authors motivate their framework by postulating that the current evaluations mostly measure linguistic quality rather than faithfulness i.e. actual reasoning structure. The paper received 4 reviews majority of which were overall positive. The rebuttal was also detailed and addressed the concerns of most of the reviewers. I recommend acceptance of the paper and request them to incorporate the rebuttals while preparing the camera ready version of the paper.